# An Artificial Magnetic Conductor-Backed Compact Wearable Antenna for Smart Watch IoT Applications

Muhammad Aamer Shahzad [1], Kashif Nisar Paracha [1], Salman Naseer [2], Sarosh Ahmad [1,3], Muhammad Malik [1,4], Muhammad Farhan [1], Adnan Ghaffar [5,*], Mousa Hussien [6,*] and Abu Bakar Sharif [1]

1 Department of Electrical Engineering and Technology, Government College University Faisalabad (GCUF), Faisalabad 38000, Pakistan; aamerx2011@gmail.com (M.A.S.); kashifnisar@gcuf.edu.pk (K.N.P.); sarosh.ahmad@alumnos.uc3m.es (S.A.); mmalik@gcuf.edu.pk (M.M.); mfarhan@gcuf.edu.pk (M.F.); abubakarsharif@gcuf.edu.pk (A.B.S.)
2 Department of Information Technology, University of the Punjab, Gujranwala Campus, Gujranwala 52250, Pakistan; salman@pugc.edu.pk
3 Department of Signal Theory and Communications, Universidad Carlos III de Madrid (UC3M), Leganés, 28911 Madrid, Spain
4 School of Micro-Nano Electronics, State Key Laboratory of Silicon Materials, Hangzhou Global Scientific, and Technological Innovation Center (HIC), ZJU-UIUC Institute, Zhejiang University, Hangzhou 310027, China
5 Department of Electrical and Electronic Engineering, Auckland University of Technology, Auckland 1010, New Zealand
6 Department of Electrical Engineering, United Arab Emirates University, Al Ain 15551, United Arab Emirates
* Correspondence: aghaffar@aut.ac.nz (A.G.); mihussien@uaeu.ac.ae (M.H.)

**Abstract:** Smart watch antenna design is challenging due to the limited available area and the contact with the human body. The strap of smart watch can be utilized effectively for integration of the antenna. In this study, an antenna integrated on a smart watch strap model using computer simulation technology (CST) was designed. The antenna was designed for industrial, scientific, and medical (ISM) frequency bands at 2.45 and 5.8 GHz. Roger 3003C was used as substrate due to its semi-flexible nature. The antenna size is $28.81 \times 19.22 \times 1.58$ mm$^3$ and it has a gain of 1.03 and 5.97 dB, and efficiency of 80% and 95%, at 2.45 and 5.8 GHz, on the smart watch strap, respectively. A unit cell was designed having a dimension of $19.19 \times 19.19 \times 1.58$ mm$^3$ to mitigate the effect of back radiation and to enhance the gain. The antenna backed by the unit cell exhibited a gain of 2.44 and 6.17 dB with efficiency of 50% and 72% at 2.45 and 5.8 GHz, respectively. The AMC-backed antenna was integrated into a smart watch strap and placed on a human tissue model to study its human proximity effects. The specific absorption rate (SAR) values were calculated to be 0.19 and 1.18 W/kg at the designed ISM frequencies, and are well below the permissible limit set by the FCC and ICINPR. Because the antenna uses flexible material for wearable applications, bending analysis was also undertaken. The indicated results prove that bending along the x- and y-axes has a negligible effect on the antenna's performance and the antenna showed excellent performance in the human proximity test. The measured results of the fabricated antenna were comparable with the simulated results. Thus, the designed antenna is compact, has high gain, and can be used effectively for wireless IoT applications.

**Keywords:** artificial magnetic conductor (AMC) plane; wearable antenna; smart watch; SAR; IoT

## 1. Introduction

In the future, the greatest challenge in networks will be to connect everything everywhere, and millions of devices will be interconnected with each other. Thus, there is a need for flexible antennas that can be integrated with flexible and portable devices. These antennas should have a low profile, be lightweight, be energy efficient, and have high gain for their effective operation [1]. For instance, the smart watch is the most widely used wearable device that can be utilized in many IoT applications [2,3]. To connect smart watches

wirelessly with other devices and sensors, a flexible and miniaturized antenna is needed to be easily integrated into IoT gadgets. A metal-rimmed antenna for a 5G/GPS-based smart watch application is proposed in [4]. Here, a shorting strip and proper feeding at an optimal location in a slotted metal frame ring provided operational bandwidths of 1575, 2500–2690, 3400–3800, and 4800–5000 MHz. A smart watch antenna with novel non-planar high impedance surfaces (HIS), instead of the tradition HIS surface, was presented and integrated in an all-metal smart watch, and showed reasonable gain and efficiency while being worn on the human wrist [5–8]. A tri-band antenna was proposed using a tempered edge and an inverted L-capacitive coupled shorted strip. The antenna showed a positive gain operating in three distinct bands with reduced specific absorption rate (SAR) values. A metamaterial-based surface was utilized to improve the gain and bandwidth, and to reduce the back radiation of the antenna. An AMC is a type of meta-material which exhibits the properties of a perfect magnetic conductor. An AMC exhibits the unique property of zero-degree phase reflection at the designed resonating frequency [9–14].

A complimentary split ring resonator (CSRR) was used to generate three different bands in [12]. The antenna showed low gain while being tested on a human tissue model. A multi-band antenna having a size of $35 \times 30 \times 0.5$ mm$^3$ was proposed in [15], using a FR-4 (4.3, 0.02) substrate. The antenna showed gain of $-1.08$, 2.4, and 1.52 dB at 2.4, 3.5, and 5.2 GHz, respectively. Another antenna having a size of $40 \times 40 \times 0.4$ mm$^3$ was designed at 1.57, 1.94, and 2.4 GHz, with gain of 0.5, 1.4 and 2 dB, respectively. The reported antenna could not be tested on a smart watch and had a large profile, and no SAR analysis was undertaken [16].

An antenna with enhanced bandwidth and compact size was proposed at 2.4 and 5.2 GHz for smart watch applications. A T-shaped structure on top of FR-4 substrate was used to obtain wider bandwidth, and higher gain and efficiency; however, SAR values were not evaluated [17]. In [18], on the top and bottom of the FR-4, two antennas were developed having dimensions of $49 \times 35 \times 5$ mm$^3$. The efficiency of the antennas ranged from 35 to 38%. The antennas had low efficiency and no simulations were performed on the human phantom model. The idea of an optically transparent dual-band antenna built on the screen of a smart watch was proposed in [19]. The antenna can operate at two resonating frequencies, namely, 2.4 and 5.2 GHz, and had efficiency of around 60%. The antenna gain was not provided and the SAR value was also not calculated, so the actual result on a smart watch was not evaluated. An antenna working at 2.4, 3.4, and 4.9 GHz frequencies were proposed in [20]. The efficiency of the antenna ranged from 67 to 91% for the proposed operating bands. Due to its good performance and simple structure, it was a good candidate for wireless portable devices; however, the antenna gain was not given, and the SAR value of this antenna was also not calculated to assess the actual result on the human tissue model. For smart watch applications, a cavity-based slot antenna was proposed in [21]. The efficiency of the antenna ranged from 57 to 66% when tested on a hand phantom model, and the SAR value of the antenna was within the permissible range set by the FCC. However, the antenna had a low gain of 1.8 dB and also a single band operating at 2.4 GHz.

A Bluetooth antenna for smart watch applications was previously proposed. It had a loop structure and was integrated with a watch's metal frame. Peak gain of 1 dB with radiation efficiency of 70% was obtained. A single layer simulation model was used to check the impact of the hand on the antenna's performance. The antenna had a lower gain of 1 dB and a large size [22]. Another tri-band antenna having a size of $35 \times 35 \times 5$ mm$^3$ was proposed and had efficiency from 76 to 86% and gain up to 1.84 dB. It was designed for the metal frame of a smart watch. The antenna efficiency, and had low gain, but it was also not tested on a watch or a human tissue model [23]. An annual ring type smart watch antenna integrated into a watch's metal frame was proposed in [24], having a gain of 4 dBi and radiation efficiency of 62% at an operating frequency 2.4 GHz. The SAR value of the designed antenna was below the permissible limit set by the FCC. A circular slot antenna operating at a single band of 2.4 GHz was designed for an all-metal smart watch

antenna, and had an efficiency of 65% and gain of 2.6 dB. The antenna was tested and simulated in free space and on a phantom hand model [25]. A dipole antenna having a size of $45 \times 25 \times 1.5$ mm$^3$ was proposed in [26]. The tri-band antenna achieved good gain. SAR values were not evaluated and the antenna was not tested on a human tissue model. A multi-band smart watch antenna was proposed in [27]. The antenna was based on a plastic material, and showed low gain and efficiency; no SAR analysis was undertaken. Another meander-shaped PIFA antenna was utilized for wrist watch applications operating at a resonant frequency of 915 MHz. As a result of the addition of a parasitic element, there was a 70% increase in the achieved bandwidth compared to the convention PIFA topology, but the antenna ha low gain and efficiency [28].

In this research, an antenna integrated on a smart watch strap model using CST was designed. The antenna was designed for ISM operating bands at 2.45 and 5.8 GHz. Roger 3003C (3, 0.0019), a semi-flex substrate, is used as the substrate of the antenna. The designed antenna, which is integrated onto a smart watch strap, has a small size of $28.81 \times 19.22 \times 1.58$ mm$^3$, and has a gain of 1.03 and 5.97 dB, and efficiency of 80 and 95%, respectively. A unit cell having a size of $19.19 \times 19.19 \times 1.58$ mm$^3$ was designed to mitigate the effect of back radiation and to increase the gain. The antenna was simulated with a unit cell and achieved the gain of 2.44 and 6.17 dB with efficiency of 50 and 72%, at 2.45 and 5.8 GHz, respectively. The SAR values of the antenna were calculated and found to be within the permissible limits set by the FCC and ICINPR, namely, 0.19 W/kg at 2.45 GHz and 1.18 W/kg at 5.8 GHz for 1 g of tissue.

The remainder of this paper is organized as follows. Section 2 presents the design analysis of the proposed antenna. The design of the unit cell is proposed in Section 3. Section 4 discusses the AMC-backed antenna and its analysis. Finally, the paper is concluded in Section 5, including a discussion of future work.

## 2. Design Analysis of the Proposed Antenna

This section explains the design analysis of the designed antenna used to accomplish the specified study goals. The schematic diagram of the proposed dual-band antenna is presented in Figure 1. The simulations and optimizations were carried out using computer simulation technology (CST) microwave studio 2018. In the first step, a simple square patch was designed, a parasitic patch on the right side of substrate was added, and then two resonators were introduced inside the ground plane on the top to resonate the antenna at 2.45 and 5.8 GHz. The antenna was printed on a Rogers 3003C flexible substrate with a dielectric constant of 3, the loss tangent of the substrate is 0.0019, and thickness used is 1.58 mm. The Roger 3003C substrate material was used because of its low profile, flexibility, and ease of fabrication and integration. The antenna has dimensions of $28.81 \times 19.22 \times 1.58$ mm$^3$. At the rear of the substrate, a ground plane with dimensions of $19.21 \times 19.22$ mm$^2$ is employed. A parasitic patch helps to improve the reflection coefficient of the antenna, hence improving the antenna's performance. The reflection coefficients were found to be $-22.9$ dB at 2.45 GHz and $-26.8$ dB at 5.8 GHz, as shown in Figure 2. The optimized parameters are given in Table 1.

### 2.1. Design Steps of the Proposed Antenna

First, a simple patch with a feed line was designed having a width of 8.64 mm and length of 6.72 mm starting from the center of the substrate (see Figure 3). A feed extension was designed having a length of 4.8 mm and width of 2.88 mm starting from the center of the edge of the substrate. Then, a parasitic element was designed having length of 17.29 mm and width of 4.8 mm. This was designed to improve the $S_{11}$ of the designed antenna to a reasonable extent. Moreover, a ground plane was created having length of 19.21 mm and width of 19.22 mm. A defective ground plane technique is used for resonance at 2.45 and 5.8 GHz frequencies. In this step, a horizontal radiator was introduced, having an initial length of 3.27 mm and a width of 1.92 mm to operate at 2.45 GHz; the other

radiator inside the ground plane had an initial length of 6.72 mm and a width of 1.92 mm to work at 5.8 GHz (see Figure 4).

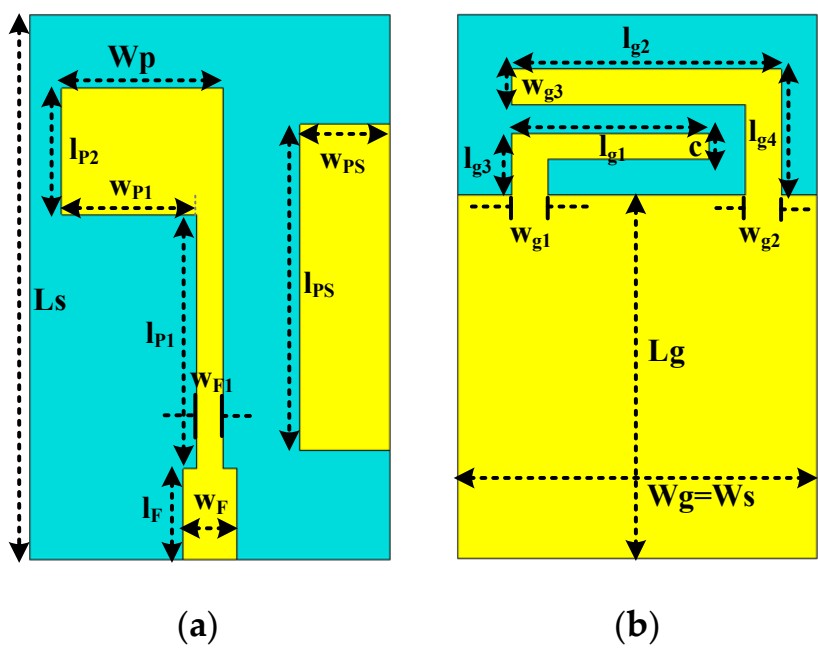

(a)                                              (b)

**Figure 1.** (**a**) Antenna front dimensions. (**b**) Antenna back dimensions.

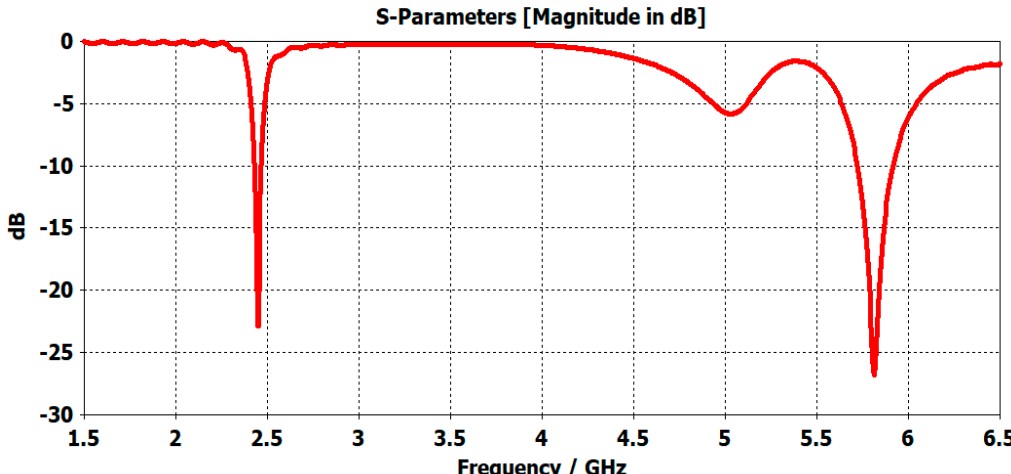

**Figure 2.** Reflection coefficient of the antenna.

**Table 1.** Proposed antenna's dimensions.

| Dimensions | Values (mm) | Dimensions | Values (mm) |
|---|---|---|---|
| Ws | 19.22 | Wg | 19.21 |
| Ls | 28.81 | Lg | 19.22 |
| Lf | 4.80 | wg1 | 1.92 |
| Wf | 2.88 | wg2 | 1.92 |
| wf1 | 1.44 | wg3 | 1.92 |
| Wp | 8.64 | lg1 | 10.56 |
| Lp | 20.17 | lg2 | 14.41 |
| wp1 | 7.20 | lg3 | 3.27 |
| lp1 | 13.45 | lg4 | 6.72 |
| Wps | 4.80 | lp2 | 6.72 |
| Lps | 17.29 | C | 1.34 |

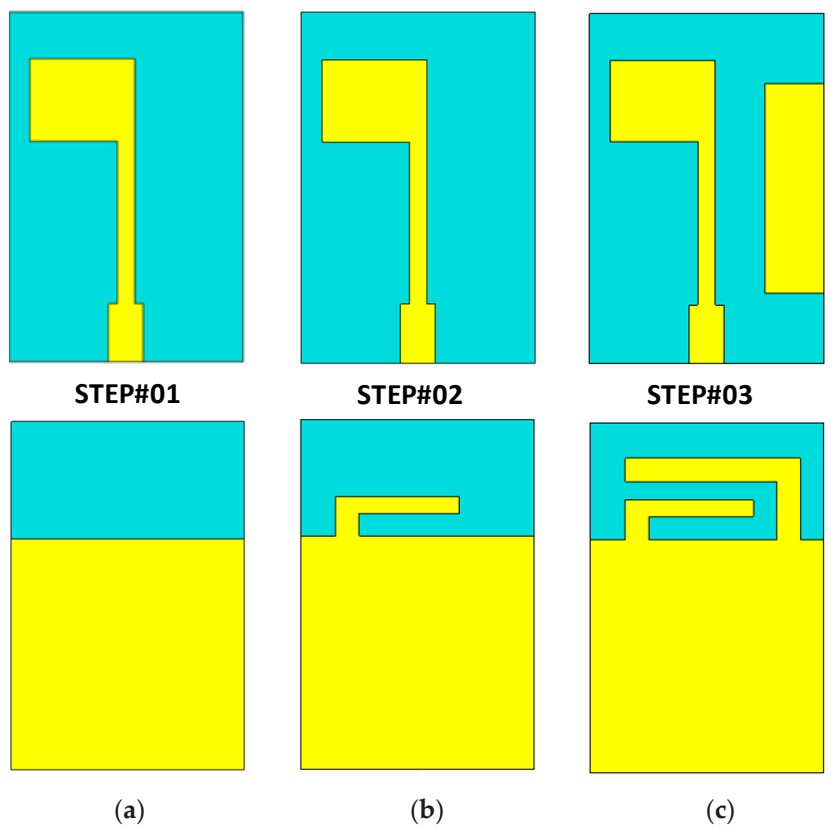

**Figure 3.** Proposed antenna design steps: (**a**) simple patch with partial ground (ANT I); (**b**) patch with one pole on the ground (ANT II); (**c**) patch with parasitic element and two poles on the ground plane (ANT III).

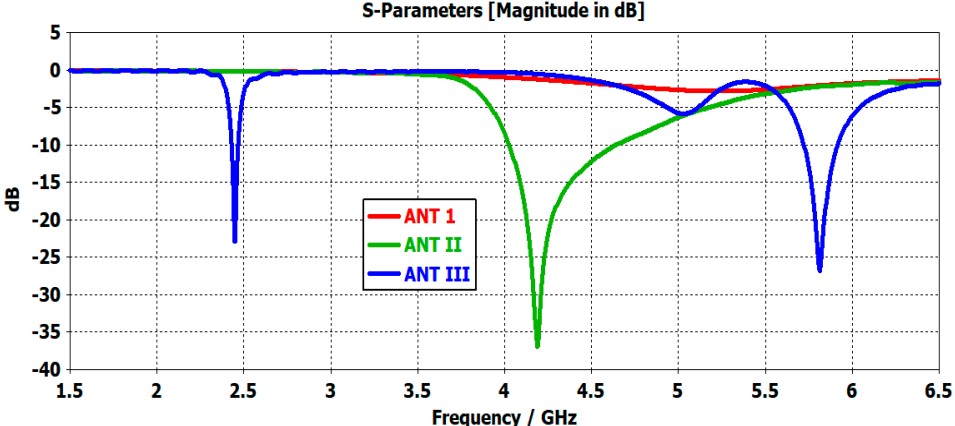

**Figure 4.** $S_{11}$ of proposed designs of the antennas (ANT I-III) in Figure 3.

## 2.2. Equivalent Circuit Model

In the advanced developed system (ADS) software®, an equivalent circuit model for the compact size dual-band antenna was designed, as presented in Figure 5. The suggested circuit model was designed for measuring the input impedance matching and can be simply used to create an equivalent circuit of a dual-band antenna. The circuit is made up of two parallel resistor–inductor–capacitor (RLC) circuits linked in series with one capacitor and one inductor. Both RLC circuits are connected in series, and the entire circuit model is made up of two resistors, three inductors, and three capacitors. In Table 2, corresponding values of each element are listed. It is evident from Figure 5a that this circuit configuration is used to achieve dual bands of 2.45 and 5.8 GHz. Resistors are used to enhance the

reflection coefficient while keeping it under an acceptable return loss. In comparison with the antenna structure, two capacitors (C1 and C2) with two inductors (L1 and L2) are used for the two radiators behind the ground plane, while the other capacitor (C0) is used for the patch with the feed line. Moreover, an inductor (L0) is used for the parasitic element and the resistors (R1 and R2) are used in each of them. The antenna's reflection coefficient may be adjusted by altering the resistor values. Figure 5b depicts the reflection coefficient of an equivalent circuit model (b).

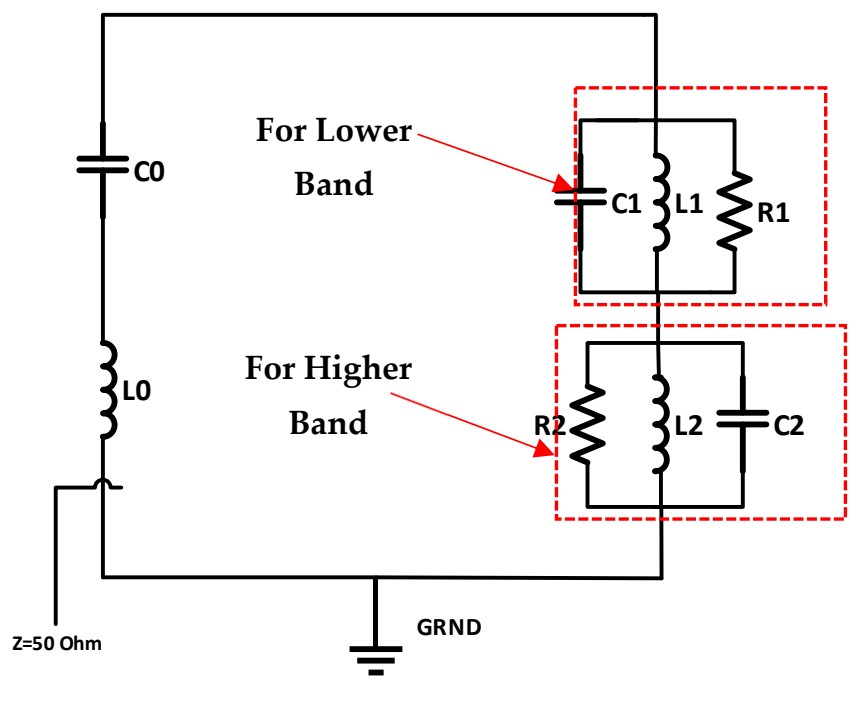

(**a**)

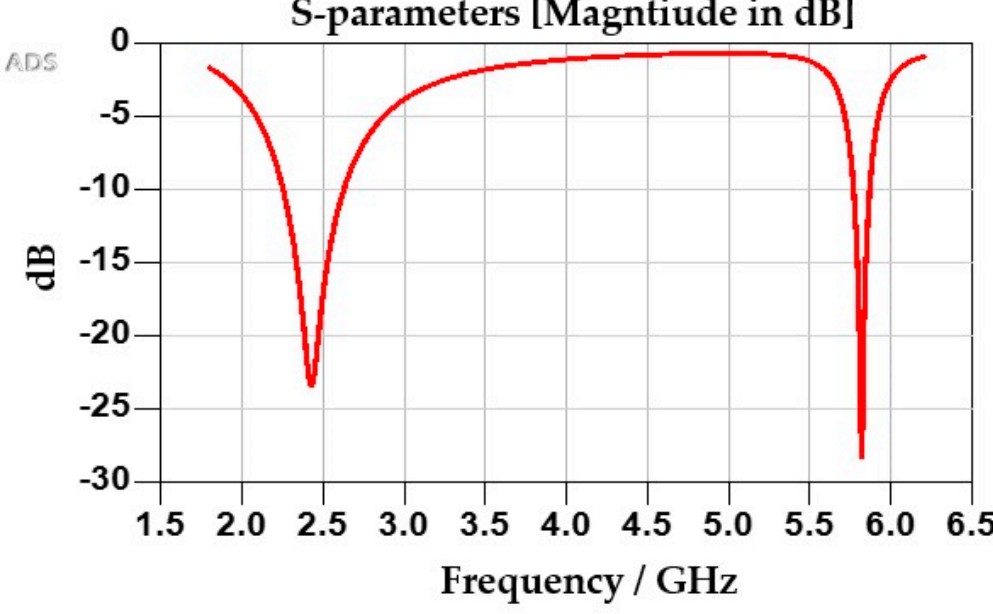

(**b**)

**Figure 5.** (**a**) Equivalent circuit model. (**b**) Reflection coefficient of equivalent circuit model.

**Table 2.** Values of the components used in the circuit model.

| Inductors | Values (*n*H) | Capacitors | Values (pF) | Resistors | Values (Ω) |
|-----------|---------------|------------|-------------|-----------|------------|
| L0 | 0.1 | C0 | 2.0 | R1 | 60 |
| L1 | 0.9 | C1 | 4.2 | R2 | 60 |
| L2 | 39 | C2 | 19 | Zs | 50 |

### 2.3. Parametric Study of the Designed Antenna

The antenna was simulated by changing the parasitic element width '*wps*' from 2 to 6 mm. The optimum results were obtained at 4.8 mm, as presented in Figure 6a. In Figure 6b, it is observed that the length of parasitic element '*lps*' was changed from 16 to 20 mm, and the optimum value was obtained at 17.29 mm. The patch's width '*wp*' was varied from 6 to 10 mm. In this case, the antenna's reflection coefficient started shifting towards the lower frequency band, as shown in Figure 6c, and the optimum value was achieved at 8.64 mm. The length of the first L-shaped radiator in the ground plane '*lg1*' was changed from 10 to 12 mm. From Figure 6e it can be observed that the best possible value at which good results were achieved at a resonating frequency was between 10 and 11 mm, achieving an optimum value at 10.56 mm. Similarly, the length of the second L-shaped radiator in the ground plane '*lg2*' was changed from 10 to 16 mm. In Figure 6f, it is observed that the best possible value at the resonating frequency was obtained between 14 and 15 mm, and the best possible results were obtained at 14.41 mm.

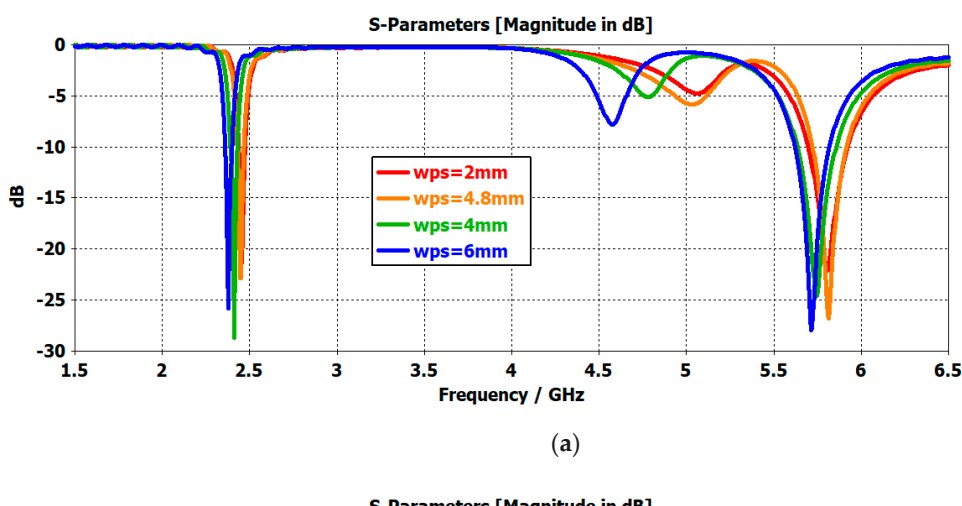

(**a**)

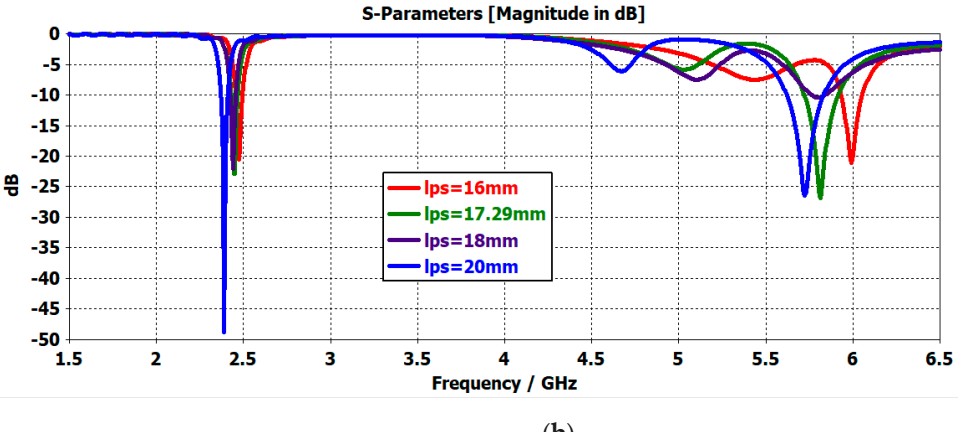

(**b**)

**Figure 6.** *Cont.*

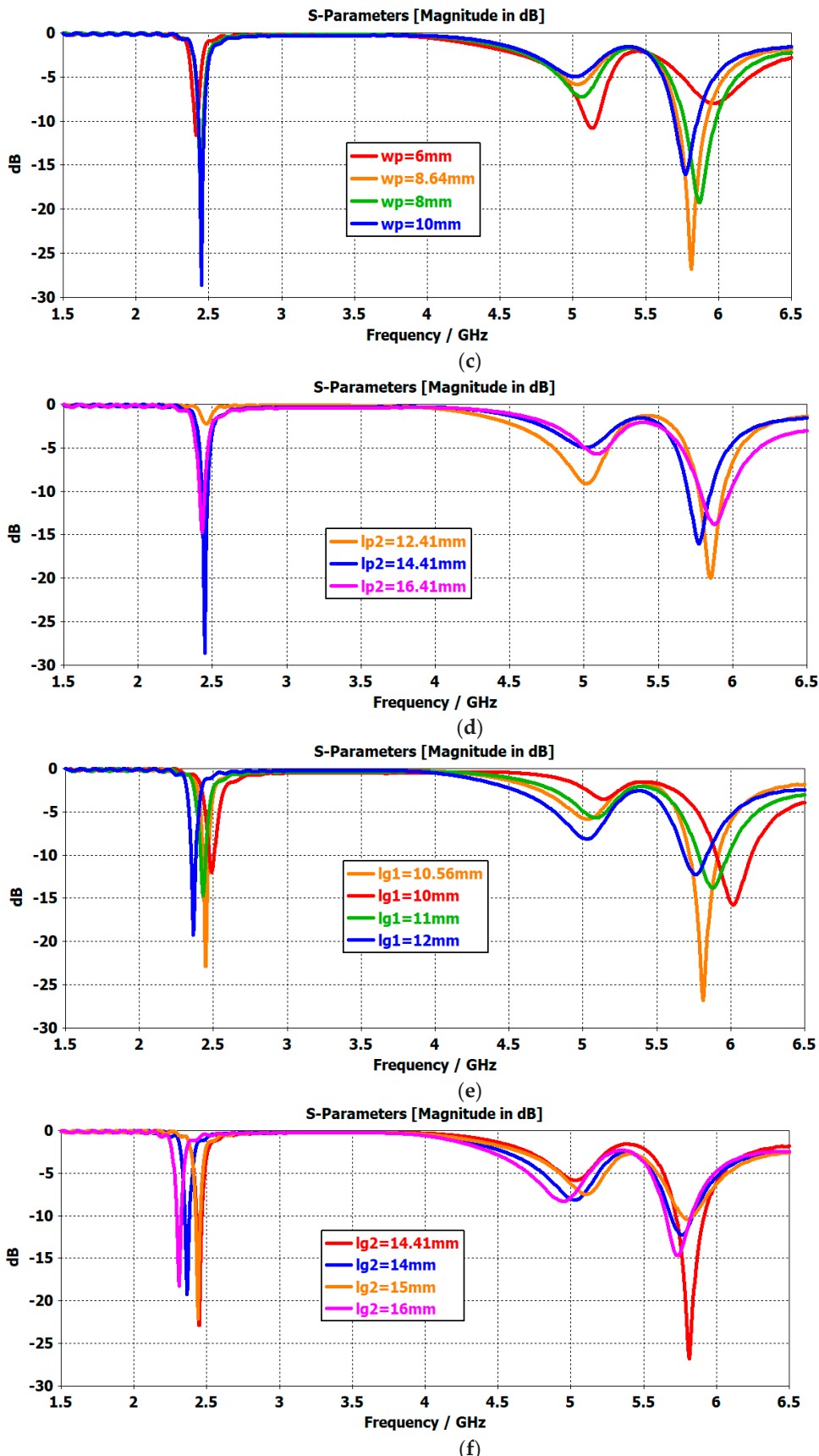

**Figure 6.** Parametric study of the antenna and the reflection coefficient [magnitude in dB] of: (**a**) "*wps*"; (**b**) "*lps*"; (**c**) "*wp*"; (**d**) "*lp2*"; (**e**) "*lg1*"; (**f**) "*lg2*".

### 2.4. Bending Analysis of the Proposed Antenna

2.4.1. Bending Analysis along the *x*-Axis

Bending analysis of the projected antenna along the *x*-axis is presented in this section. Figure 7a presents the bending radius of the antenna along the *x*-axis termed as "*Bx*". Moreover, the bending analysis was analyzed using different values of "*Bx*", such as 30, 60, and 90 mm. From the graph, it is clear that the reflection coefficient of the antenna was stable even when it was bent along the *x*-axis by 90 mm. Bending along the *x*-axis had an almost negligible effect on the lower band. However, a minor impact was seen on the higher band, as a slight shift in the $S_{11}$ curve towards the resonating frequency below 5.8 GHz; this was considered inconsequential. Overall, the bending effect is almost negligible and the proposed antenna is well suited for a wearable and flexible electronics application.

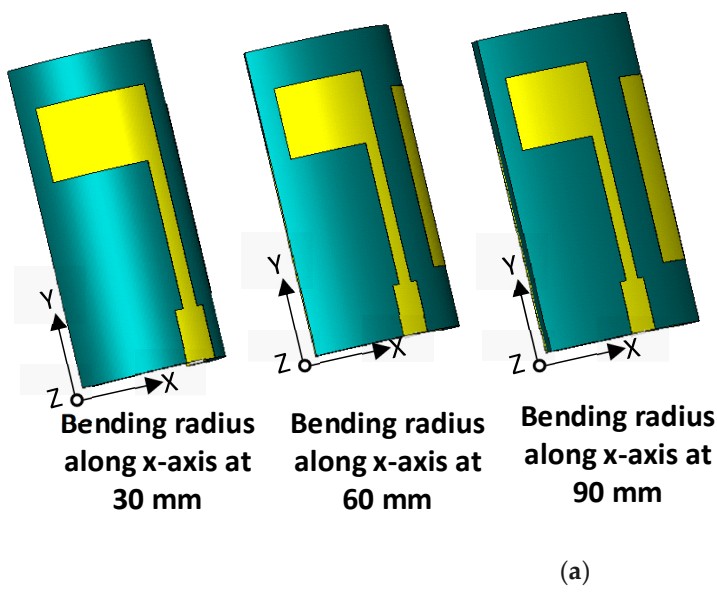

**Bending radius along x-axis at 30 mm**  **Bending radius along x-axis at 60 mm**  **Bending radius along x-axis at 90 mm**

(**a**)

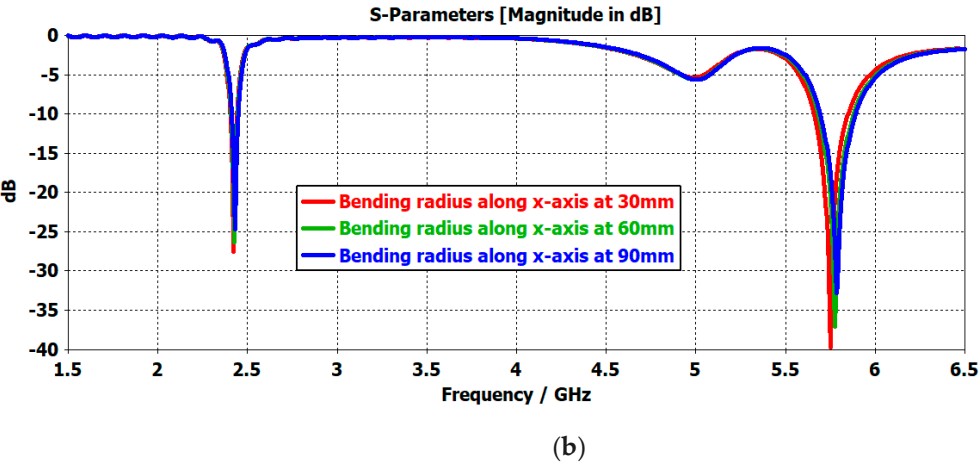

(**b**)

**Figure 7.** (**a**) Bending analysis of the antenna along the *x*-axis; (**b**) $S_{11}$ behavior at *Bx* = 30, 60 and 90 mm.

2.4.2. Bending Analysis along the *y*-Axis

This section presents the bending radius of the antenna along the *y*-axis termed as "*By*". Bending values along the *y*-axis were varied from 30 to 90 mm in order to observe bending along the *y*-axis. It can be seen that bending along the *y*-axis has a negligible effect on the antenna reflection coefficient, implying that our antenna is resistant to changes in bending along the *y*-axis (see Figure 8).

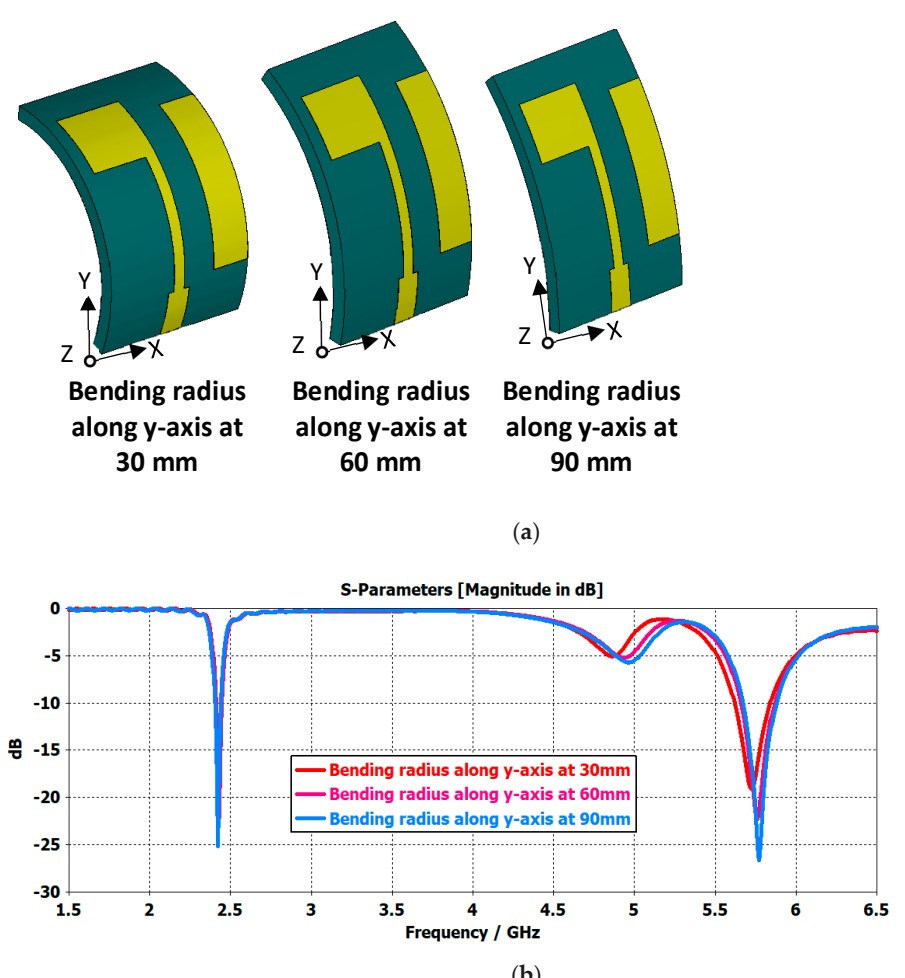

**Figure 8.** (**a**) Bending analysis of the antenna along the *y*-axis; (**b**) $S_{11}$ behavior at $By$ = 30, 60, and 90 mm.

### 2.5. Radiation Pattern of Proposed Antenna

Figure 9 depicts the proposed antenna's far-field emission pattern at 2.45 GHz. At 2.45 GHz, the antenna gains along the E- and H- planes are −0.26 and 1.03 dB, respectively. Figure 9 depicts the built antenna radiation pattern at the resonance frequency of 5.8 GHz (b). The antenna gains in the E- and H- planes are 4.97 and 4.29 dB, respectively, which are comparable with those of previously designed antennas.

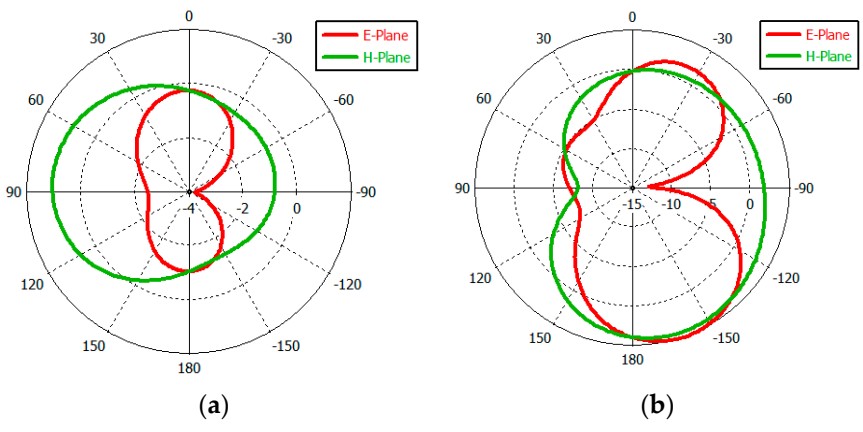

**Figure 9.** Simulated two-dimensional radiation pattern (**a**) at 2.45 GHz; (**b**) at 5.8 GHz.

## *2.6. Surface Current Density of the Proposed Antenna*

The surface current density of an antenna indicates which part of the antenna is playing a significant role in making it resonate at the desired frequency. The current distribution of the antenna without the AMC-plane is shown in Figure 10. From Figure 10, it can be clearly seen that at a lower frequency band (i.e., 2.45 GHz), most of the current flows through the back radiating elements inside the ground plane (see Figure 10a). In contrast, at the higher frequency band (i.e., 5.8 GHz), most of the current flows through the parasitic element on the top of the substrate, the feed line, and the defected ground plane, as shown in Figure 10b.

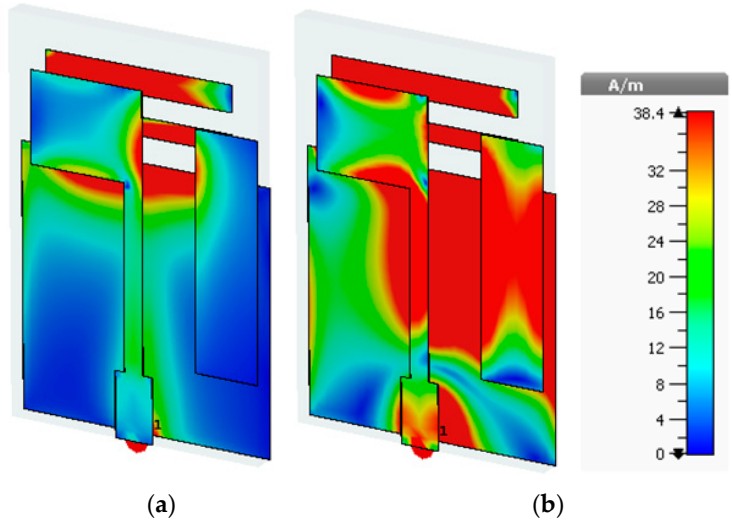

(**a**)                (**b**)

**Figure 10.** Surface current distribution (**a**) at 2.45 GHz; (**b**) at 5.8 GHz.

## 3. Design of an AMC Unit Cell

An artificial magnetic conductor (AMC) can be used to control the propagation of electromagnetic waves, which makes them suitable to improve gain and ensure a directional radiation pattern [12]. A simple and miniaturized AMC unit cell was designed on a Roger 3003C substrate (keeping the thickness of the substrate (ts) equal to 1.58 mm and the tangent loss (tanδ) equal to 0.002), with volumetric dimensions of $19.22 \times 19.22 \times 1.58$ mm$^3$, as shown in Figure 11. Moreover, its optimized dimensions are summarized in Table 3 and its zero-reflection phase at 2.45 and 5.8 GHz can be seen in Figure 12. A parametric study of the unit cell is also presented to understand its design procedures.

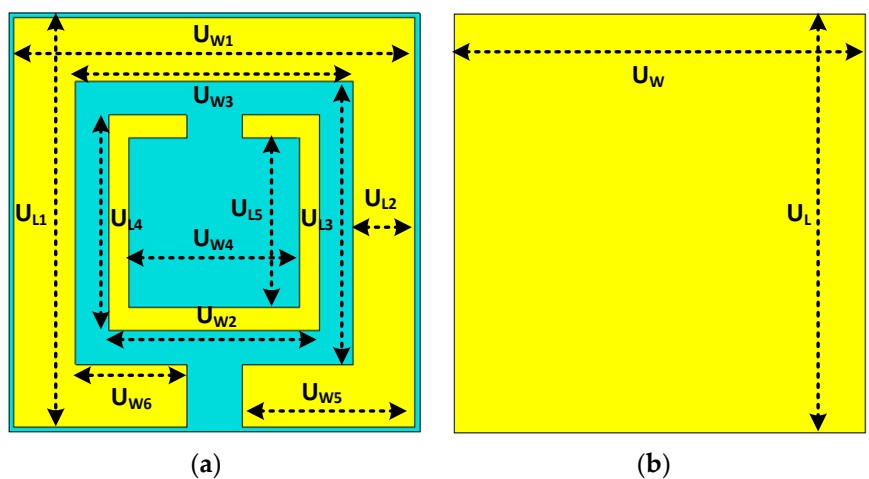

(**a**)                (**b**)

**Figure 11.** Unit cell dimensions: (**a**) front dimensions, (**b**) back dimensions.

**Table 3.** Dimensions of the unit cell.

| Dimensions | Values (mm) | Dimensions | Values (mm) |
|---|---|---|---|
| Uw | 19.22 | Ul | 19.22 |
| Uw1 | 18.79 | Ul1 | 18.79 |
| Ul2 | 2.90 | Ul3 | 13 |
| Ul4 | 9.87 | Ul5 | 7.79 |
| Uw2 | 9.87 | Uw3 | 13 |
| Uw4 | 7.79 | Uw5 | 8.10 |
| Uw6 | 5.20 | | |

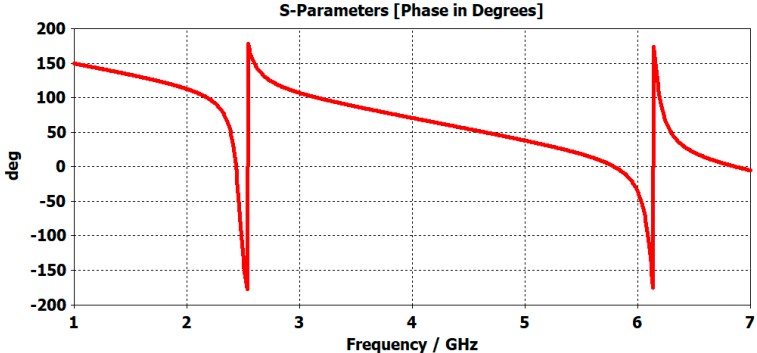

**Figure 12.** Zero-phase reflection coefficient of the unit cell.

As the first step, Figure 13a illustrates the design of a simple square patch (AMC I). It can be observed that this unit cell provided zero-phase reflection at 2.8 GHz, as illustrated in Figure 13b. In the next step, a square slotted patch (AMC II) was designed to make a dual reflection phase. It can be seen in Figure 13b that this modification in design produced zero-reflection phases at 2.5 and 6.6 GHz. Then, in the third step, an inner square copper patch with slots on the upper side was introduced (AMC III). To accomplish the objective, a second modification resulted in a zero-reflection phase at 2.45 and 5.8 GHz, as illustrated in Figure 13b. This unit cell exhibited a zero-reflection phase at 2.45 and 5.8 GHz in a miniaturized form to operate as a dual-band reflector for the proposed antenna. Simulated reflection phases for the different design steps are summarized in Figure 13b.

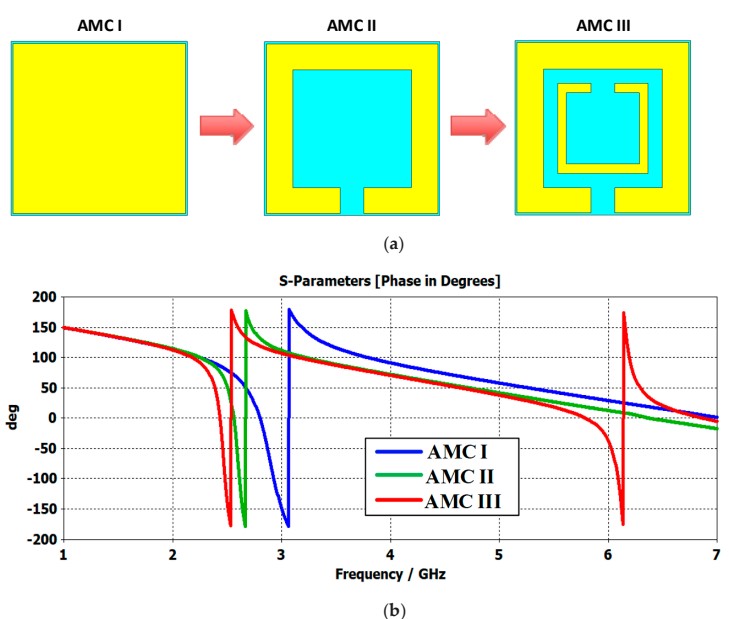

**Figure 13.** Unit cell (**a**) design steps; (**b**) unit cell phase.

### 3.1. Surface Current Distribution

The surface current distribution of the unit cell is presented in Figure 14. The surface current density shows that the outer slots play a significant role in this unit cell, having phase zero at 2.45 GHz (see Figure 14a). Figure 14b illustrates that the current distribution in the inner square patch is greater than in any other region, implying that this portion is crucial in ensuring that this unit cell has zero-phase difference at 5.8 GHz.

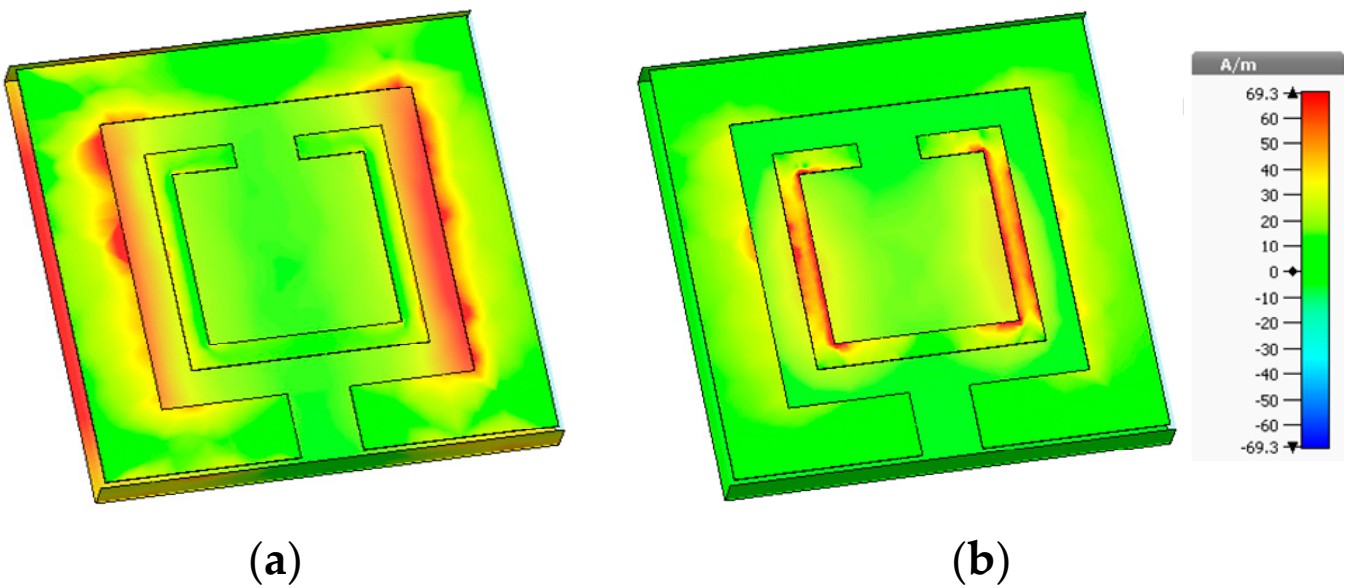

**Figure 14.** Surface current distribution of the AMC unit cell at: (**a**) 2.45 GHz; (**b**) 5.8 GHz.

### 3.2. Parametric Analysis of the AMC Unit Cell

Three key parameters of the AMC unit cell are studied in this section: the length of the inner square patch '*ul4*', the lower width of the outer square patch '*uw5*' and its upper width '*uw3*'. Figure 15a illustrates that when length '*ul4*' was changed from 8.87 to 10.87 mm, the zero-phase difference does not change at the upper frequency band. However, it slightly changes in the lower band as it starts moving towards the frequency of 3 GHz when the length is increased, and the optimum value is obtained at 9.87 mm. Thus, this value is taken as the length of the inner patch in the design of the unit cell. When '*uw5*', the lower width of the outer patch, was changed from 7.10 to 9.10 mm, it can be seen from the parametric graph that a slight shift is observed; the phase difference at the lower band shifts below the desired frequency and at the upper band shifts beyond the desired frequency. An optimum value of 8.10 mm was used for this design, at which better results were obtained.

### 3.3. Design of an AMC-Backed Antenna

In order to decrease the potential entanglement of the design with human bodies, an AMC is employed as the back plane. Figure 1a depicts the proposed AMC-backed antenna configuration, where the proposed antenna is maintained at a short distance "*D*" from the AMC plane. The AMC plane works as an electro-magnetic reflector. The AMC is placed behind the substrate at a distance of 10 mm. By changing the values of "*D*", the difference in the $|S_{11}|$ values are presented in the Figure 16c. From the graph, it can be noted that the reflection coefficient of the antenna is enhanced by increasing the distance "*D*" from the AMC plane. Practically, a foam material having the electrical properties of air is used to replace the air gap. The foam is placed between the antenna and the AMC, and has a thickness of 10 mm, as presented in Figure 16b for practical demonstration.

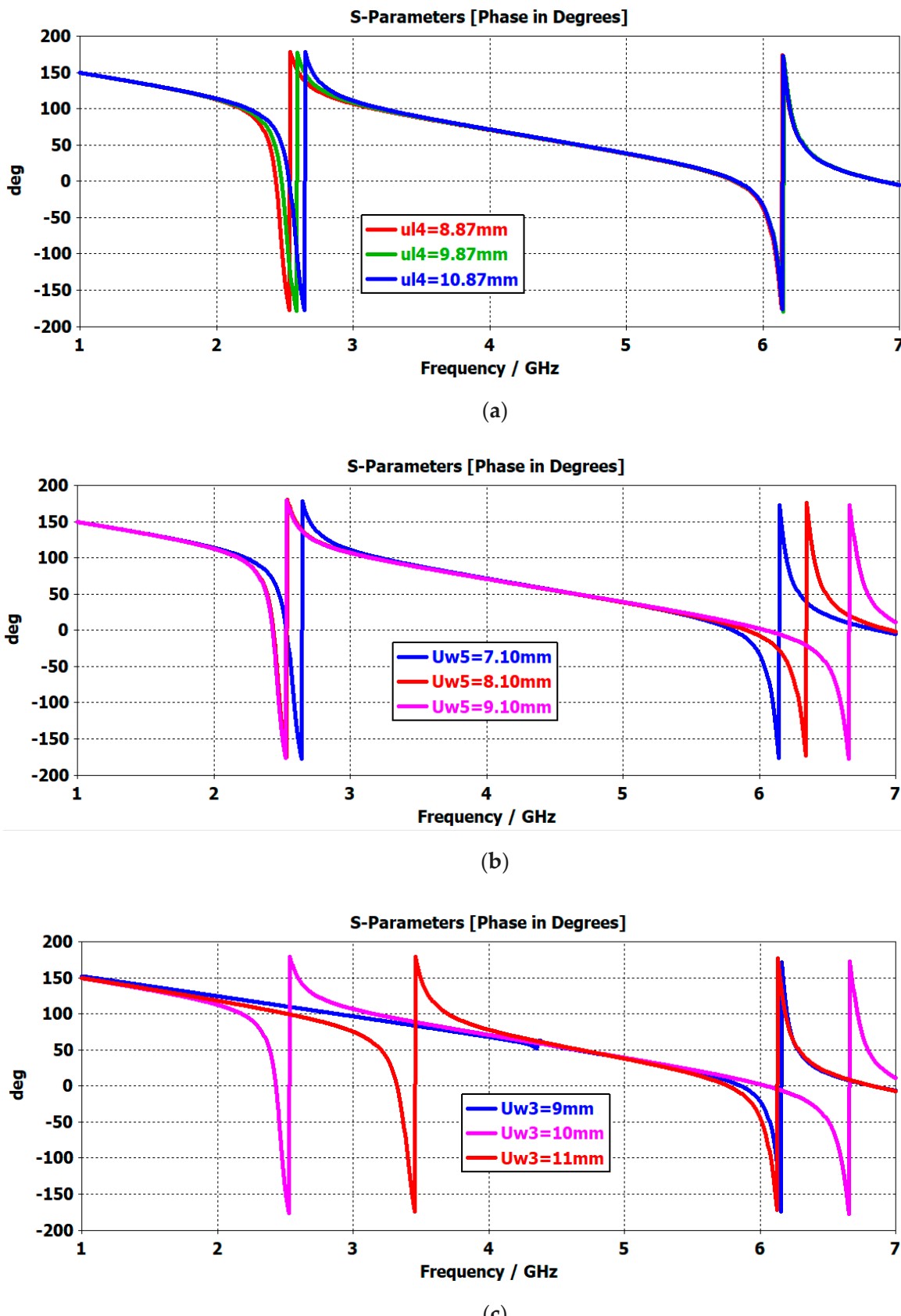

**Figure 15.** Parametric study of the unit cell: (**a**) variation in "*ul4*"; (**b**) variation in "*uw5*"; (**c**) variation in "*uw3*".

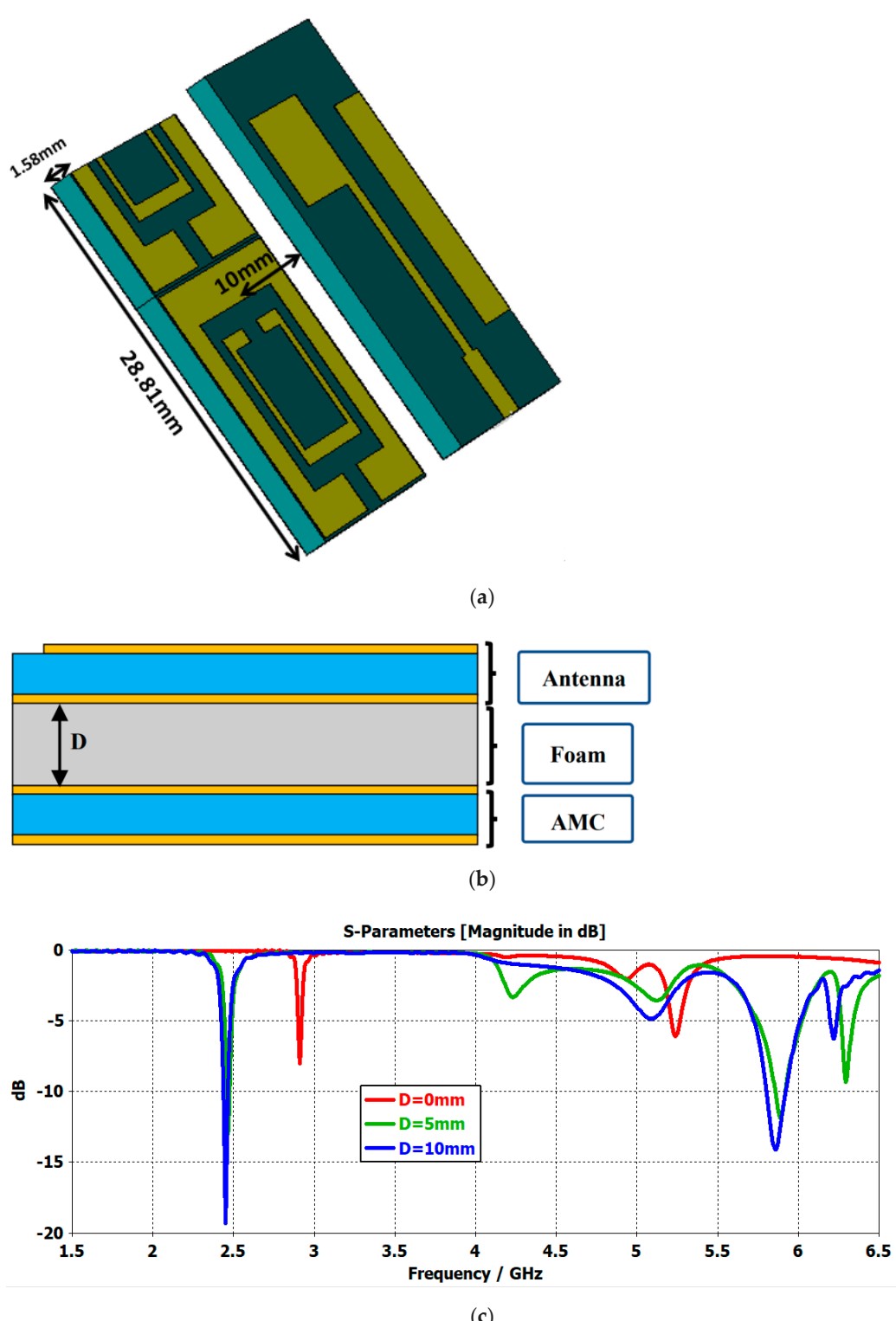

**Figure 16.** Designed antenna with an AMC at the back: (**a**) antenna with the AMC; (**b**) foam covering cap between the AMC and the antenna; (**c**) $S_{11}$ at different distances from the AMC.

Figure 17 depicts the surface current distribution of an AMC-backed antenna. At 2.45 GHz, the current mostly travels via the patch and the defective ground plane of the AMC-backed antenna. In the higher frequency band (5.8 GHz), some current passes through the parasitic patch, the defective ground plane, and the reflector behind the antenna.

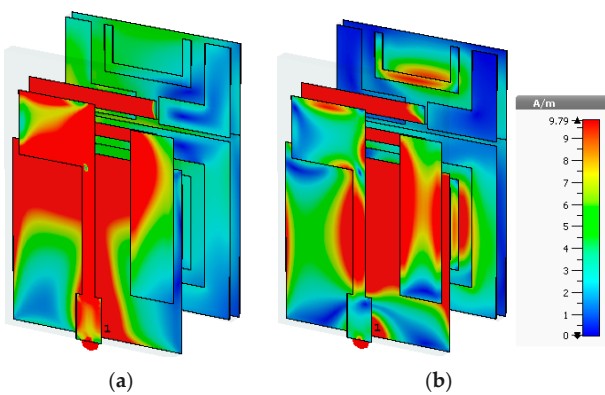

**Figure 17.** Surface current distribution of the AMC-backed antenna: (**a**) at 2.45 GHz, (**b**) at 5.8 GHz.

### 3.4. Bending Analysis of the AMC-Backed Antenna

3.4.1. Bending Analysis along the *x*-Axis

The bending analysis of the AMC-backed antenna along the *x*-axis is presented in this section. Figure 18a presents the bending analysis antenna along the *x*-axis termed as "*Bx*". The bending analysis was analyzed by using different values of "*Bx*", such as 30, 60, and 90 mm, as can be seen in Figure 18. From the graph, it is clear that the reflection coefficient of the antenna is stable even if it is bent along the *x*-axis by 90 mm. Bending along the *x*-axis has an almost negligible effect on lower band. However, a minor impact was seen on the higher band, as a slight shift in the $S_{11}$ curve towards the resonating frequency below 5.8 GHz; this was considered inconsequential. Overall, the bending effect was almost negligible and proposed antenna is well suited for a wearable and flexible electronics application.

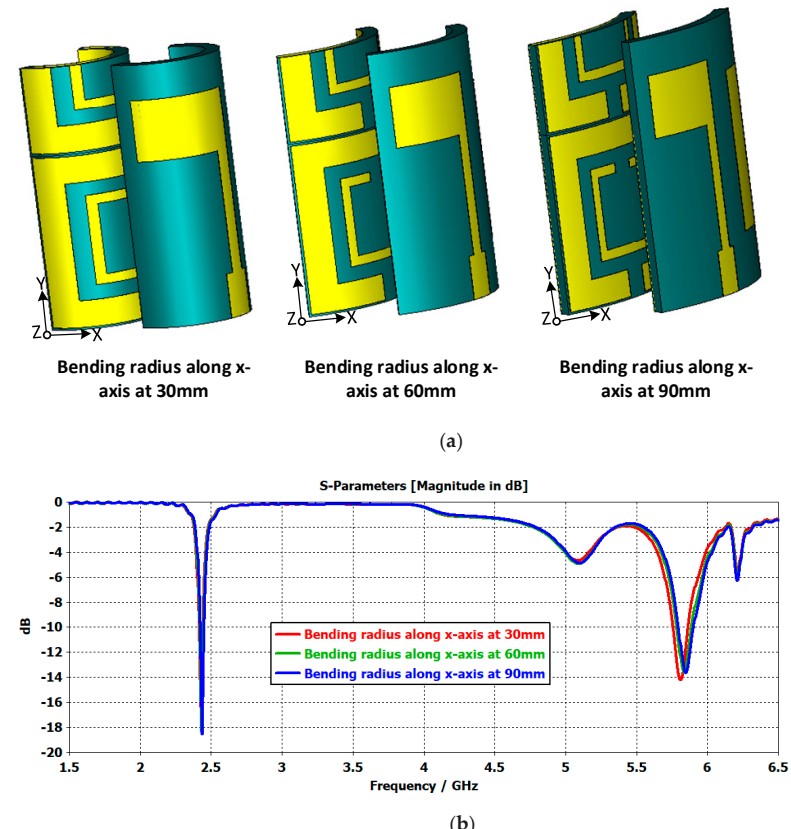

**Figure 18.** (**a**) Bending analysis of the AMC-backed antenna along the *x*-axis. (**b**) $S_{11}$ behavior at Bx = 30, 60, and 90 mm.

### 3.4.2. Bending Analysis along $y$-Axis

This section presents a bending study of the AMC-backed proposed antenna along the $y$-axis. In order to observe bending along the $y$-axis, the bending values were varied in a range (i.e., 30–90 mm) along the $y$-axis. It can be seen that bending along the $y$-axis has a negligible effect on the antenna reflection coefficient, implying that our antenna is resistant to changes in bending along the $y$-axis (see Figure 19).

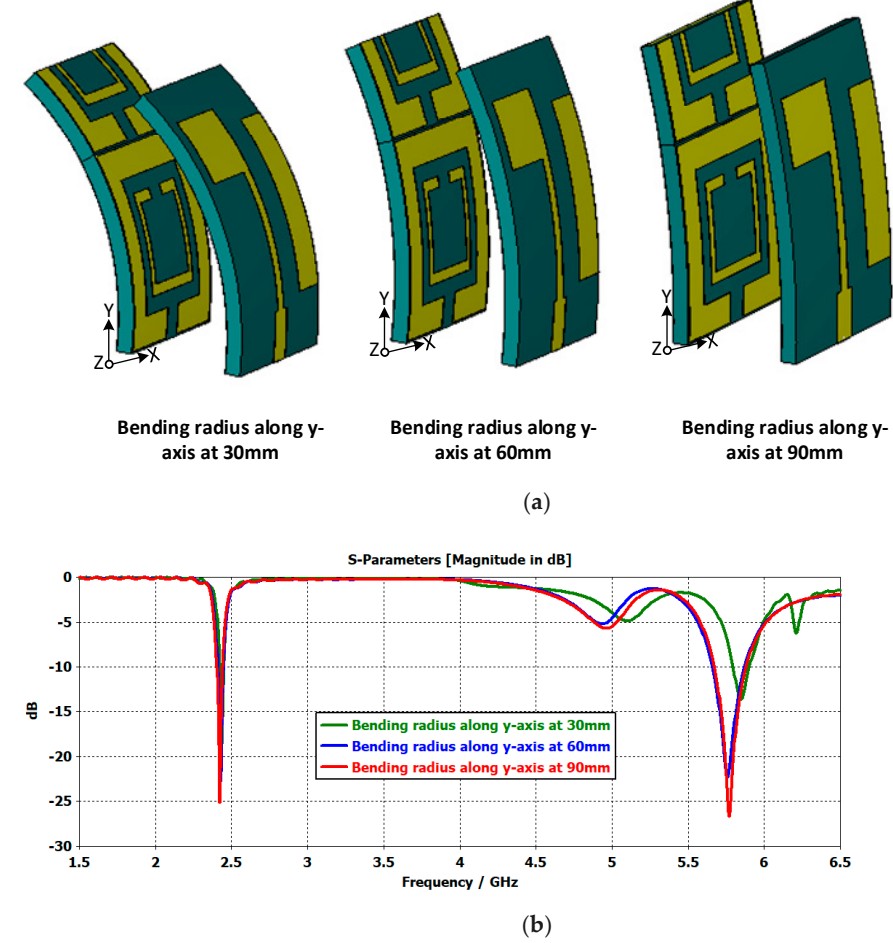

**Figure 19. (a)** Bending analysis of the AMC-backed antenna along the $y$-axis; **(b)** $S_{11}$ behavior at By = 30, 60, and 90 mm.

## 4. Specific Absorption Rate (SAR) Analysis

Specific absorption is the amount of radio frequency energy that is absorbed by human tissue when it is released. It is determined by taking an average over a given volume of 1 or 10 g. The SAR limit in the United States is 1.6 W/kg for 1 g of tissue, whereas in Europe it is 2 W/kg for 10 g of tissue [21].

The expression for the relationship between the input power and the SAR is evaluated as follows [23]:

$$\text{SAR} = \frac{\sigma |E^2|}{\rho} \tag{1}$$

where '$\sigma$' and '$\rho$' denote the electrical conductivity (S/m) and the mass density (kg/m$^3$), respectively, and '$E$' is the electric field intensity (V/m). The electric power intensity is related to the signal power and is evaluated as follows:

$$\text{Power} \left( \frac{\text{W}}{\text{m}^2} \right) = \frac{(\text{E(V/m)})^2}{377} \tag{2}$$

SAR values of the antenna were measured at 2.45 and 5.8 GHz. The SAR value is important, as is depicts the effect of back radiation on human tissue. If it is more than the prescribed limit set by the FCC and ICINPR, then it will damage the tissue of the human body. The SAR values were calculated and found to be 0.95 W/kg at 2.45 GHz and 1.56 W/kg at 5.8 GHz for 1 g of tissue (see Figure 20), which is less than but close to the standard limits. However, to further decrease the SAR values, an AMC backing was utilized. With an AMC backing, it can be seen that the SAR value was calculated to be 0.19 W/kg at 2.45 GHz and 1.18 W/kg at 5.8 GHz for 1 g of tissue (see Figure 21). Thus, with the input power of 0.5 W, the SAR values of our antenna are within the acceptable range for both bands.

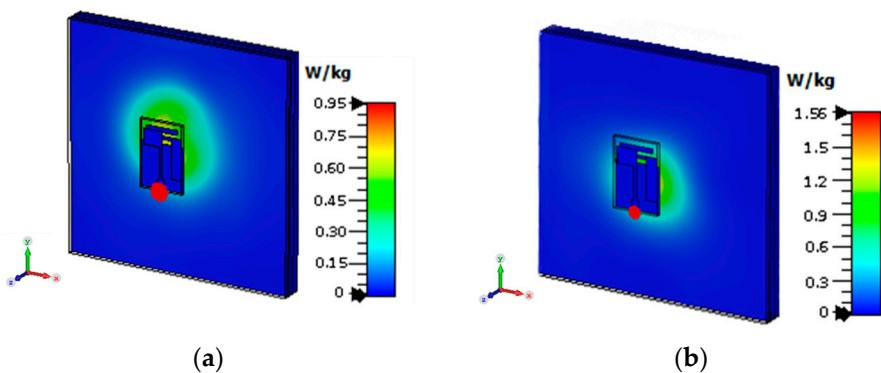

(**a**)                    (**b**)

**Figure 20.** SAR of the antenna: (**a**) at 2.45 GHz; (**b**) at 5.8 GHz.

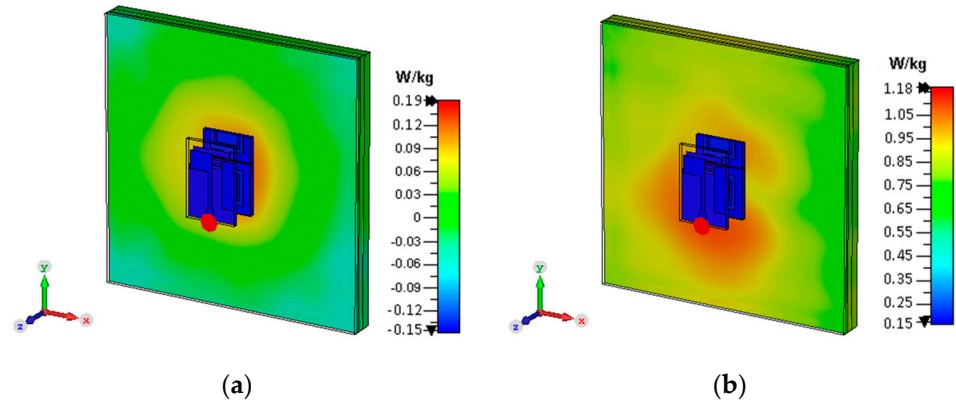

(**a**)                    (**b**)

**Figure 21.** SAR of the AMC-backed antenna: (**a**) at 2.45 GHz; (**b**) at 5.8 GHz.

## 5. Fabrication and Measurements

### 5.1. Fabrication and Measurements of the Proposed Antenna

Figure 22 shows the front and rear views of the fabricated prototype of the proposed dual-band patch antenna, which was built on a flexible Roger 3003C substrate. In the case of the simulated results, the antenna reflection coefficient covers the impedance bandwidth from 2.4 to 2.46 GHz (1.55%) at 2.45 GHz, and from 5.68 to 5.88 GHz (3.44%) at 5.8 GHz. In the case of the measured results, the antenna covers the bandwidth from 2.39 to 2.43 GHz (1.6%) at 2.45 GHz, and from 5.64 to 5.85 GHz (3.6%) at 5.8 GHz (see Figure 23). The slight deviation between the simulated and measured results is due to some unavoidable artifacts during fabrication. The antennas comprised a semi-flexible material formed on a Roger 3003C substrate in a technology process comprising the following basic stages. First, a copper annealed conductive coating was deposited onto the Rogers substrate [29,30]. In the second stage, the pattern of the antenna structure was made on the copper film using a laser inkjet printing technique. In the third stage, the substrate was cut into individual samples of specified sizes. The thickness of the copper coating was measured to be about 0.035 mm. The coating had a uniform thickness, without cracks or defects. No detachment

from the substrate was observed [31,32]. Fabricated examples of flexible antennas for ISM bands 2.45 and 5.8 GHz are shown in Figure 22.

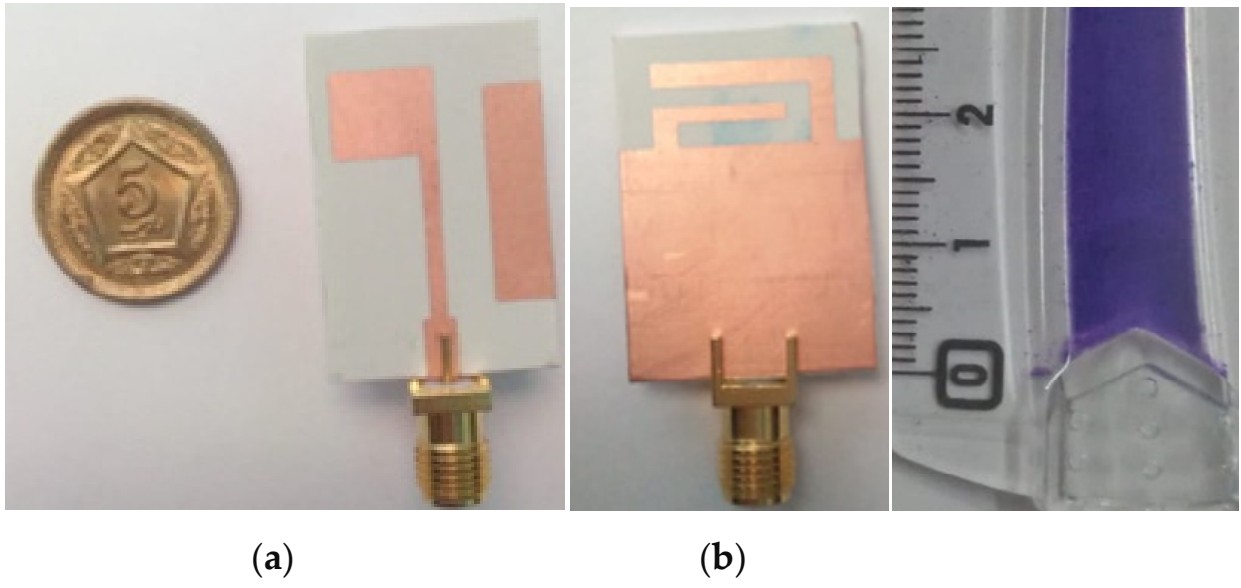

**(a)**        **(b)**

**Figure 22.** Fabricated prototype of the proposed antenna: (**a**) front view; (**b**) back view.

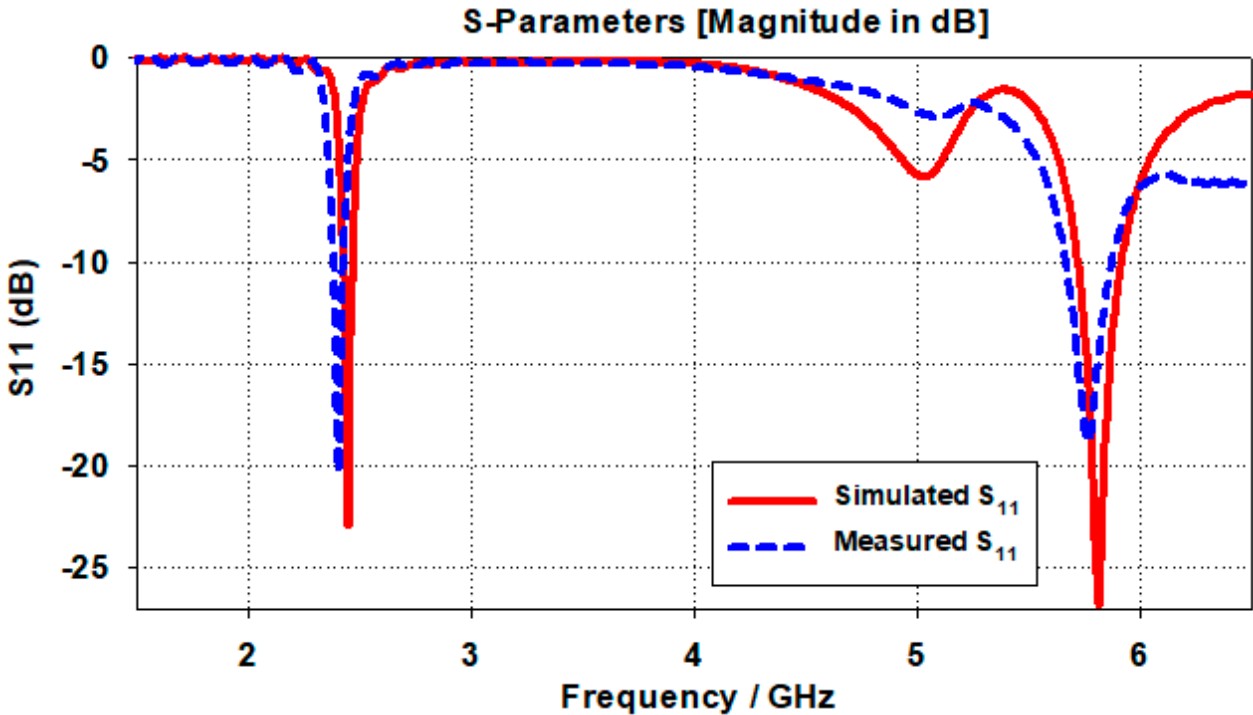

**Figure 23.** Simulated and measured reflection coefficient ($S_{11}$(dB)) of the proposed antenna.

*5.2. Fabrication and Measurements of the AMC-Backed Antenna in Free Space*

Figure 24 compares the simulated and measured reflection coefficients of the proposed dual-band antenna with and without the AMC plane, demonstrating their consistency. At 2.45 and 5.8 GHz, the simulated reflection coefficient is near to 20 and 15 dB, respectively. In free space, the AMC-backed antenna has reflection coefficients of −15 and −20 dB operating at 2.45 and 5.8 GHz, respectively. The simulated and measured results were found to be consistent with the proposed response of the antenna.

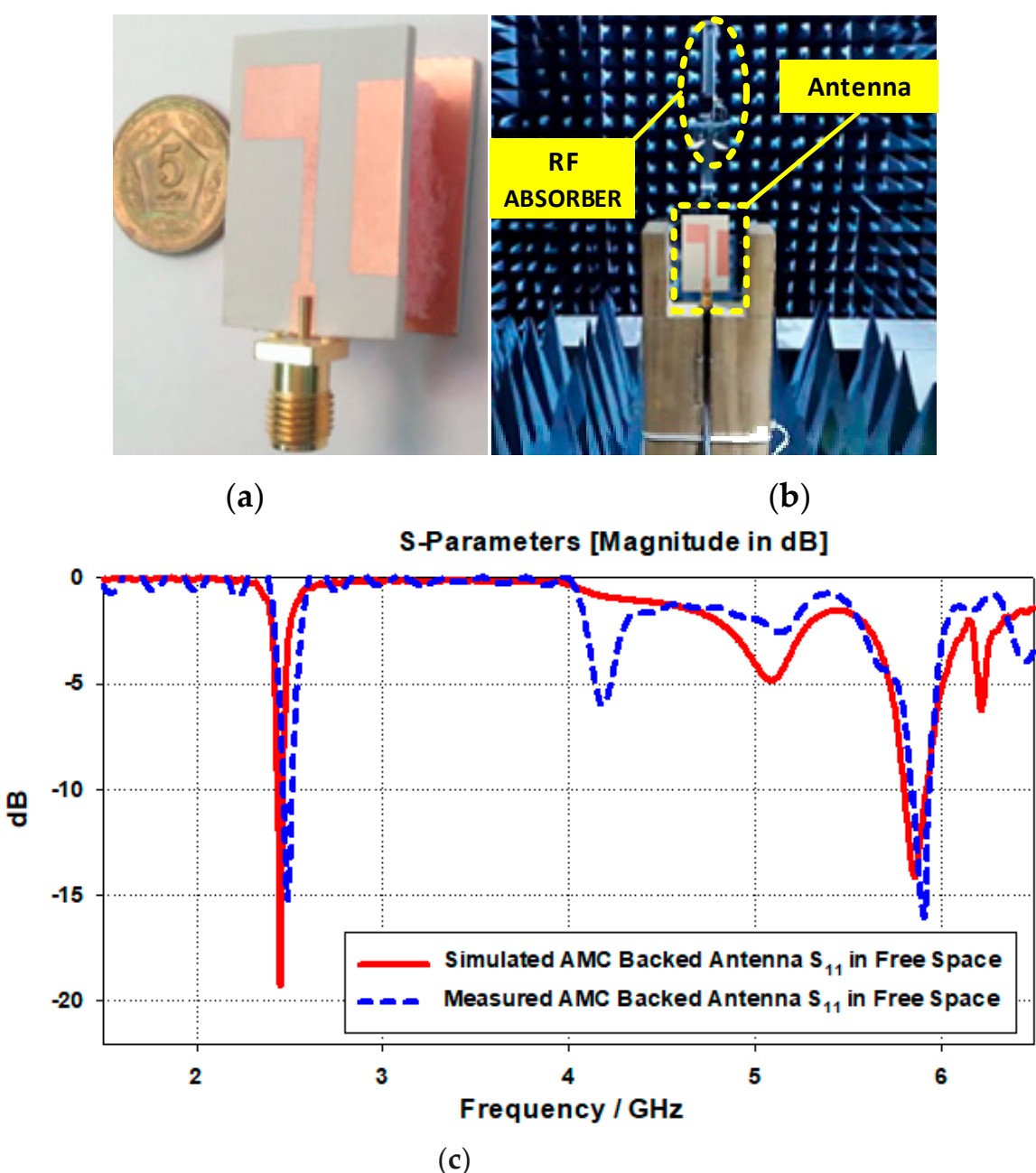

**Figure 24.** (**a**) Fabricated prototype of the AMC-backed antenna; (**b**) evaluation setup in an anechoic chamber; (**c**) comparison of measured and simulated reflection coefficients of the AMC-backed antenna in free space.

Figure 25 depicts the antenna simulated radiation pattern represented by a solid black line, whereas the antenna measured results are represented by a dotted red line with the unit cell in free space. As can be seen from the pattern, the radiation patterns are almost identical, implying that our fabricated antenna produces nearly the same results as the proposed design of the simulated antenna.

### 5.3. Fabrication and Measurements of the AMC-Backed Antenna on a Smart Hand Watch

Ideally, wearable antennas must be designed to avoid significant effects of coupling from human tissues. A human tissue model consisting of skin, fat, and muscle was used to assess the robustness of the designed antenna when it comes into contact with human tissue. The skin had a thickness of about 1 mm, an epsilon of 41.3, and electrical conductivity of 0.88 S/m; fat had a thickness of 3 mm, an epsilon value of 5.3, and electrical conductivity

of 0.05 S/m; and the muscle tissue had a thickness of 4 mm with permittivity of 54.8 and electrical conductivity of 0.96 S/m, as given in Table 4. The rubber strap of the smart watch had a thickness of 2 mm (see Figure 26).

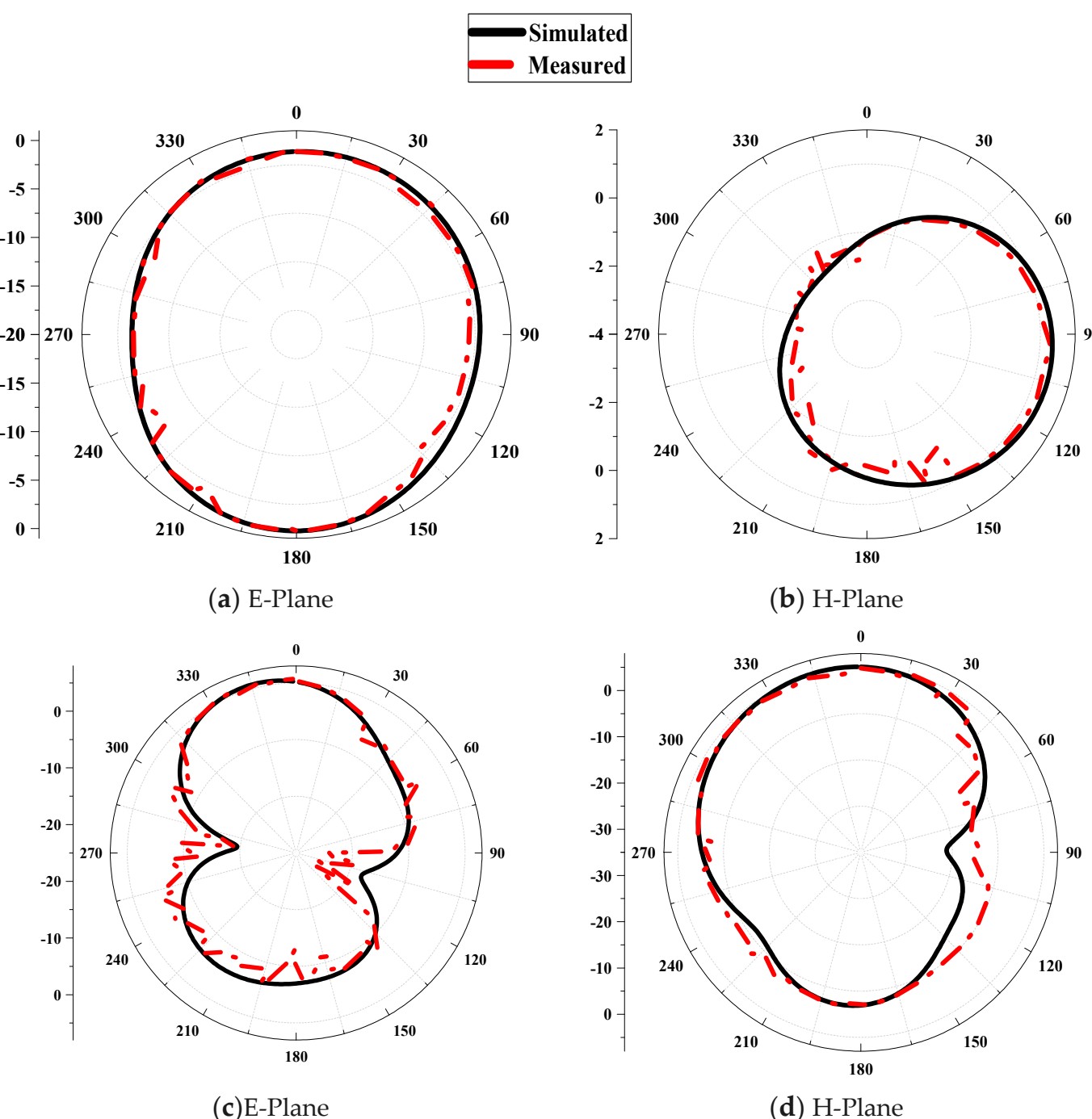

**Figure 25.** Simulated and measured 2D radiation patterns of the AMC-backed antenna in free space: (**a**) at 2.45 GHz (E plane); (**b**) at 2.45 GHz (H plane); (**c**) at 5.8 GHz (E plane); (**d**) at 5.8 GHz (H plane).

**Table 4.** Properties of different human tissue layers [19].

| Human Tissues | Relative Permittivity | Electrical Conductivity (S/m) | Thickness (mm) |
|---|---|---|---|
| Skin | 41.3 | 0.88 | 1 |
| Fat | 5.3 | 0.05 | 3 |
| Muscle | 54.8 | 0.96 | 4 |

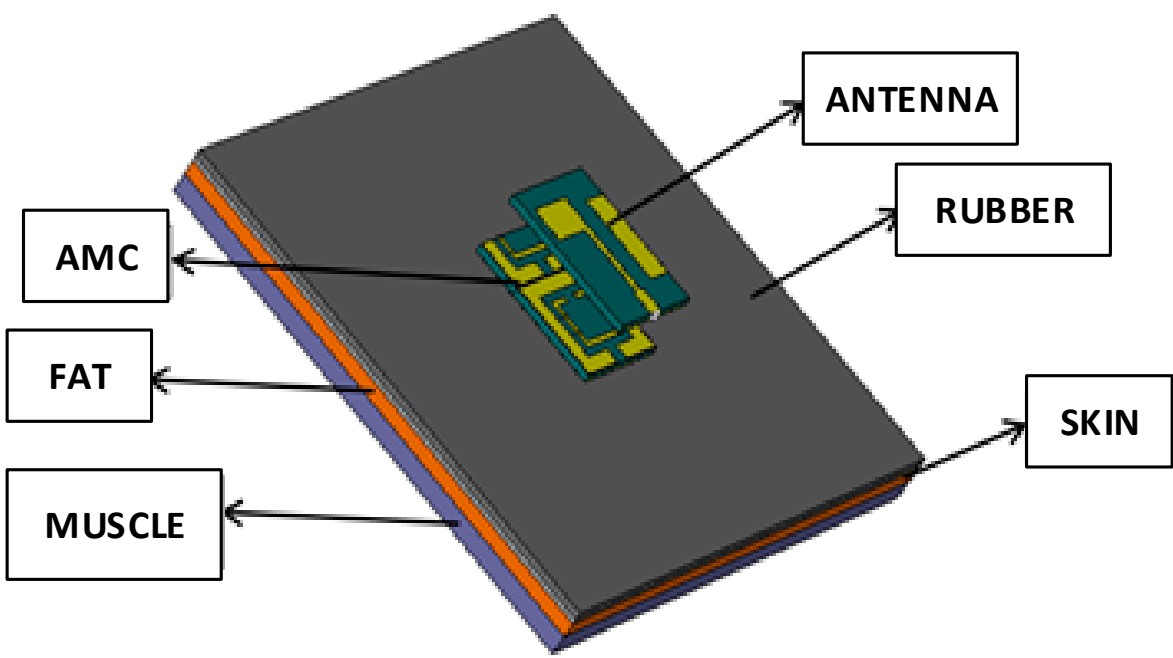

**Figure 26.** Three-dimensional layer model of human tissues with the AMC-backed antenna.

Figure 27 depicts the simulated and measured reflection coefficients of the AMC-backed antenna on the human phantom. Simulations on the human phantom revealed that the AMC-backed antenna had a reflection coefficient close to −14 dB at 2.45 GHz for both simulated and measured outcomes, and close to −16 dB at 5.8 GHz for both simulated and measured results. This implies that the simulated and measured plots can be regarded as a good match.

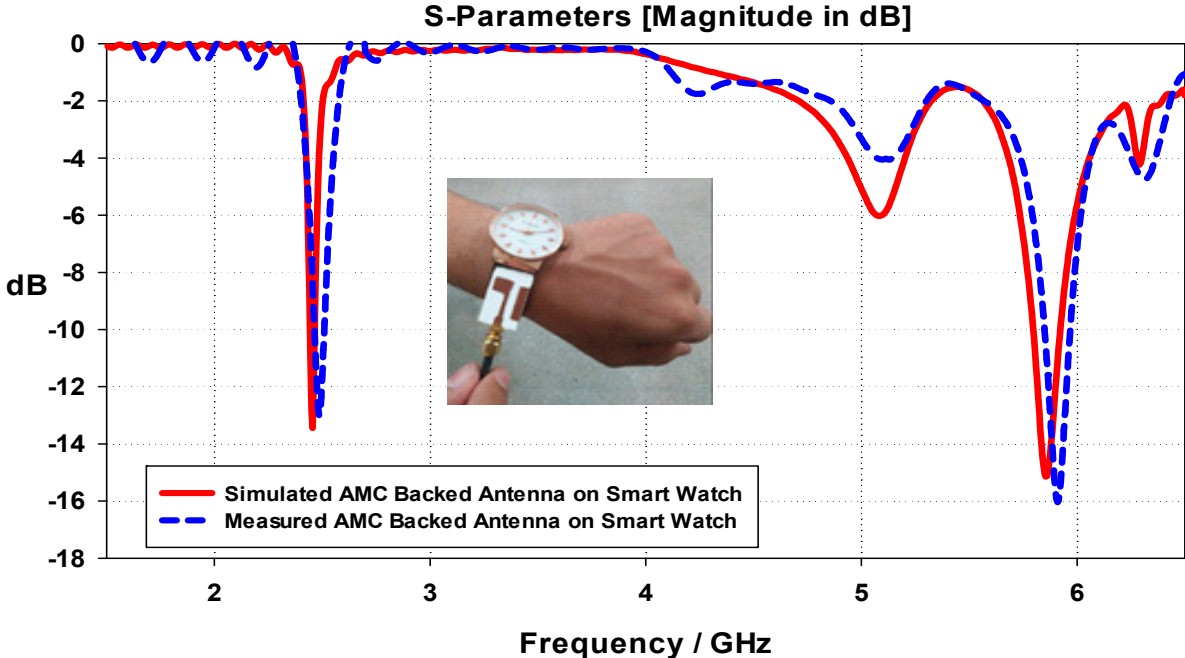

**Figure 27.** Comparison of simulated and measured reflection coefficients of the AMC-backed antenna on a human phantom.

Figure 28 illustrates the simulated and measured radiation patterns of the proposed AMC-backed antenna on the E- and H-planes on the watch and human hand. The patterns indicate that the antenna is directional at both 2.45 and 5.8 GHz; thus, the antenna has

almost negligible back radiation. Simulations also showed on-hand realized gains of 2.44 and 2.27 dB at 2.45 GHz, and 6.17 and 5.62 dB at 5.8 GHz. The total radiation measured efficiency was found to be more than 50% when the AMC-backed antenna was placed over human tissues. The simulated and measured radiation patterns were almost identical to each other; thus, the designed antenna can be used practically and implemented for wireless data communication, and provides good gain, radiation pattern, and efficiency. A comparison of the antenna with an AMC backing with previous research is given in Table 5.

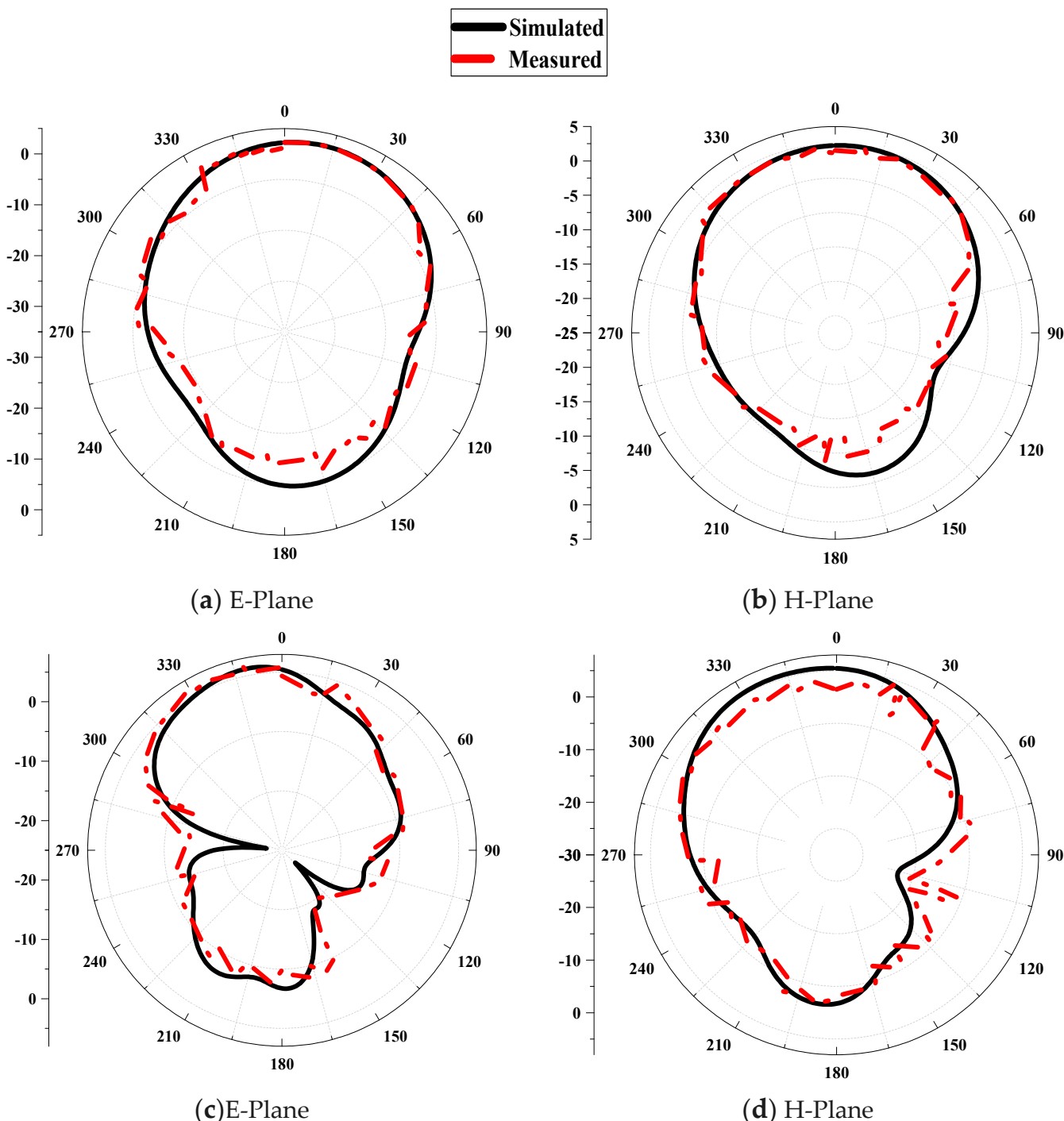

**Figure 28.** Simulated and measured 2D radiation patterns of the AMC-backed antenna on a watch plus human tissue: (**a**) at 2.45 GHz (E plane); (**b**) at 2.45 GHz (H plane); (**c**) at 5.8 GHz (E plane); (**d**) at 5.8 GHz (H plane).

**Table 5.** Comparison of this work with previous research.

| Ref. No. | Dimensions (mm$^3$) | Operating Frequency (GHz) | Substrate Material | Realized Gain (dB) | Total Efficiency (%) | SAR (W/kg) @ 1 g |
|---|---|---|---|---|---|---|
| [8] | $40 \times 124 \times 1.2$ | 0.7, 1.8 | Rubber | <3.8 | - | 0.18 and 1.26 |
| [9] | $40 \times 38 \times 0.5$ | 2.47 | Metal | <3.76 | <95 | - |
| [10] | $33 \times 41 \times 1.2$ | 1.5, 2.4 | Polycarbonate | 2.2 and 4.4 | 63, 83 | 0.021 |
| [11] | $42 \times 32 \times 1.6$ | 0.63, 3.21, 3.63 | FR-4 | <3.69 | - | - |
| [12] | $31 \times 30 \times 1.6$ | 2.4, 3.5 | FR-4 | <2.25 | - | - |
| [13] | $30 \times 15 \times 1.6$ | 2.4 | FR-4 | <4 | - | - |
| [14] | $35 \times 32 \times 1.6$ | 1.9, 2.3, 2.4, 2.6, 5.2, 5.8 | FR-4 | <6.6 | <82 | - |
| [15] | $35 \times 30 \times 0.5$ | 1.7, 2.4, 3.5, 5.1 | FR-4 | <2.04 | 96, 94, 95, 54 | - |
| [16] | $40 \times 40 \times 0.4$ | 1.57, 1.94, 2.4 | FR-4 | 0.5, 1.4, 2 | 53, 89, 90 | - |
| [17] | $30 \times 30 \times 1$ | 2.4, 5.2 | FR-4 | - | - | - |
| [18] | $49 \times 35 \times 5$ | 1.68, 1.8, 2.4 | FR-4 | <1.57 | <38 | - |
| [19] | $38 \times 32 \times 1.125$ | 2.4, 5.2 | Display glass | - | 60, 65 | - |
| [20] | $32 \times 32 \times 0.4$ | 2.4, 3.4, 4.9 | FR-4 | | 68, 91, 74 | - |
| [21] | $(3.14 \times 21^2 \times 10$ | 2.4 | Rogers4050 | 1.84 | <66 | 0.515 |
| [22] | $50 \times 40 \times 5$ | 2.4 | FR-4 | 1 | <67 | - |
| [23] | $35 \times 35 \times 5$ | 1.57, 2.4, 3.5 | FR-4 | 0.84, 1.54, 1.8 | 75, 86, 86 | - |
| [24] | $37.5 \times 37.5 \times 7.25$ | 2.4 | FR-4 (4.3, 0.02) | <2.48 | <62 | 1.0569 |
| [25] | $(3.14 \times 23^2 \times 10)$ | 2.4 | - | <2.6 | <65 | - |
| [26] | $45 \times 25 \times 1.5$ | 0.8, 2.55, 3.5 | - | <4.7 | <88 | - |
| [27] | $40 \times 40 \times 1.52$ | 0.9, 1.9, 2.5, 1.5, 2.4, 5 | Plastic | 0.42, 1.53, 1.79, 1.17, 3.16, 1.63 | <50 | - |
| [28] | $43.5 \times 28.5 \times 1.2$ | 0.915 | Plastic | −0.77 | <46 | 0.004 |
| **[This Work]** | **$28.81 \times 19.22 \times 1.58$** | **2.45, 5.8** | **Roger 3003C** | **2.44, 6.17** | **50, 72** | **0.19 and 1.18** |

## 6. Conclusions

A dual-band AMC-backed miniaturized antenna was designed for ISM frequency bands of 2.45 and 5.8 GHz. Roger 3003C (3, 0.0019) is used as a substrate to utilize its flexibility. The proposed antenna was designed with smaller dimensions of $28.81 \times 19.22 \times 1.58$ mm$^3$. The antenna demonstrated an almost identical performance on a smart watch strap. A unit cell was designed having a size of $19.19 \times 19.19 \times 1.58$ mm$^3$ to mitigate the effect of back radiation and to increase gain. The antenna's SAR value was tested and found to be within the FCC and ICINPR acceptable limits to ensure that the proposed antenna is safe to be used as wearable device. Because the antenna was designed to be wearable, the effect of bending was also evaluated and found to be an insignificant influence on antenna performance. The antenna is compact and has high gain, making it suitable for wireless data transfer and wearable electronics. The SAR values were calculated to be 0.19 and 1.18 W/kg at the designed ISM frequencies, and are less than the limits set by the FCC and ICINPR. Results of the bended analysis proved that bending along the *x*- and *y*-axes had a negligible effect on the antenna's performance, and the antenna showed excellent performance in the test of human proximity. The measured results of the fabricated antenna were comparable with the simulated results. Furthermore, the antenna achieved good measurement results and is a perfect candidate for smart watch wireless IoT applications. The antenna can be used to wirelessly transmit and receive data in wearable applications.

**Author Contributions:** Conceptualization, M.A.S., S.N. and S.A.; methodology, M.A.S., S.N., K.N.P.; software, M.A.S., S.A.; validation, K.N.P., M.M., S.N., A.B.S. and A.G.; formal analysis, S.A., M.F. and A.G.; investigation, M.A.S., A.B.S., K.N.P., M.M.; resources, S.A., K.N.P., M.F.; data curation, M.A.S., S.A. and A.G.; writing—original draft preparation, M.A.S., S.A. and A.G.; writing—review and editing, K.N.P., A.B.S., M.F. and M.H.; supervision, M.H., K.N.P., M.M.; funding acquisition, M.H. and A.G. All authors have read and agreed to the published version of the manuscript.

**Funding:** This research was partially funded by United Arab Emirates University, Al Ain, UAE and the University of the Punjab, Lahore, Pakistan.

**Conflicts of Interest:** The authors declare no conflict of interest.

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
