# Peer review of "An Artificial Magnetic Conductor-Backed Compact Wearable Antenna for Smart Watch IoT Applications"

_electronics, doi:10.3390/electronics10232908_

Round 1

Reviewer 1 Report

The topic is timely and exciting, especially the usage of AMC for wearable devices; however, the authors are suggested to improve the quality of the paper from the following points:

1) The proposed antenna is intended to be used in IoT applications. Can the authors integrate it with any intended system? What will be the impact on the performance when combined with the device?

2)  What is the input power of the SAR analysis?

3) The authors should briefly explain how the equivalent circuit model (Figure 5) relating to the antenna design (Figure 3).  For instance, in terms of its slots, defective ground plane, etc., with the R, L, C components.

4) For bending analysis, it is recommended that the author analyze the antenna integrated with the AMC because the idea of this work is that the smartwatch integrates with AMC.

5) Figure 13. It is recommended to rephrase the labeling to AMC instead of Ant

6) Table  4 - please state the reference

7) The proposed antenna is quite large, and most smartwatch devices include a few other sensors such as heart rate, SpO2, etc. How does the proposed antenna integrate into modern miniaturized devices?

8) Table 5 - the overall dimension on the antenna, including the AMC, should be 8.81×19.22×1.58 463 mm3, not 19.22×28.81×1.58 m3 (AMC unit cell)

8) Conclusions and future work should be extended to contain practical applications based on the research title of this paper.

Author Response

We would like to thank Reviewer 1 for the careful review and most valuable comments that helped in improving the quality of the manuscript. We have carefully evaluated these comments, now enumerated in the attached file together with our responses.

Reviewer 2 Report

The article presents interesting results of the development, fabrication, and study of the flexible antenna for industrial, scientific, and medical (ISM) frequency bands. The topic is quite essential for flexible electronics. But in presented form article needs a lot of significant improvements. Several detailed comments are listed below:

  1. Page 6, line 185: In subsection 2.3 authors consider the parametric study of design antenna. “The optimal results are obtained at 4.8 mm that are presented in Fig. 6(a).” But in Fig. 6(a) results for only 2 mm, 4 mm, and 6 mm were presented. The results for the wps=4.8 mm are absent. Therefore authors should include the dependences of S11 versus frequency corresponding to the optimal value of “wps” (Fig.6a). The same thing corresponds to illustrating the dependences of S11 versus frequency for to the optimal value of “lps” (Fig.6b), “lg1” (Fig.6e), “lg2” (Fig.6f).
  2. To my mind, the caption for Fig.6 should be changed: it is better to name it as “Reflection coefficient [Magnitude in dB]” or “Reflection losses [Magnitude in dB]”
  3. Fig. 7(a) should be remade because it is absolutely unclear what is the “Bx”. Authors also should clearly name and describe the “Bx” parameter: it is bend radius of bend diameter or something else... The same suggestion corresponds to the Fig. 8(a) and “By” parameter.
  4. Page 10, lines 251-254. I think the explanation of the influence of surface current distribution on the operation of the antenna is very poor and unclear. On the one hand, Fig. 10 (a) shows that both "upper" and "lower" radiators are red (the maximum value of surface current) at 2.45 GHz and therefore it seems that they both play a major role in operation at 2.45 GHz. On the other hand, one can see that at 5.8 GHz both "upper" and "lower" radiators are only a little bit colored in red color. Thus, the presented Fig. 10 does not answer the question about the roles of the "upper" and "lower" radiators at 2.45 GHz and at 5.8 GHz. Moreover, the presented explanation is in contradiction with the results presented in Fig3 and Fig.4. Authors should carry out more detailed studies about the roles of the "upper" and "lower" radiators at 2.45 GHz and at 5.8 GHz.
  5. Page 11, line 259: In my opinion, “promulgation” is not a suitable term, maybe “propagation” will be the better.
  6. Page 16, line 351: Formula for SAR calculation should carefully be remade (see, for example, an instance in Wikipedia).
  7. Authors should consider the fabrication process of the antenna with more details. Also, the used fabrication process should be compared with other different approaches described in:

1) Godlinski D., Zichner R., Zöllmer V., Baumann R. Printing technologies for the manufacturing of passive microwave components: antennas // IET Microwaves, Antennas & Propagation. 2017. Vol. 11, â„– 14. P. 2010–2015.

2) Starodubov, A.V., Galushka, V.V., Serdobintsev, A.A., Pavlov, A.M., Korshunova, G.A., Ryabukho P.V., Gorodkov, S.Yu. “A Novel Approach for Fabrication of Flexible Antennas for Biomedical Applications,” 2018 18th Mediterranean Microwave Symposium (MMS), Istanbul, pp. 303-306 (2018)

3) Mohamadzade, B.; Hashmi, R.M.; Simorangkir, R.B.V.B.; Gharaei, R.; Ur Rehman, S.; Abbasi, Q.H. Recent Advances in Fabrication Methods for Flexible Antennas in Wearable Devices: State of the Art. Sensors 2019, 19, 2312. https://doi.org/10.3390/s19102312

4) Starodubov A. V. Serdobintsev A. A., Kozhevnikov I. V., Galushka V. V., Pavlov A. M. Laser ablation and other manufacturing approaches for flexible antenna fabrication // Saratov Fall Meeting 2019: Laser Physics, Photonic Technologies, and Molecular Modeling / ed. Derbov V.L. SPIE, 2020. Vol. 1145804, â„– April 2020. P. 40.

  1. Pages 19-20, lines 418-421 and Table 4: Thermal conductivity is not measured in Siemens per meter (S/m) values. I think here the authors mean electrical conductivity instead of thermal conductivity. Or I am wrong?
  2. The authors should explain why they have chosen Roger 3003C substrate as a flexible substrate.

Author Response

We would like to thank Reviewer 2 for the careful review and most valuable comments that helped in improving the quality of the manuscript. We have carefully evaluated these comments, now enumerated in the attached PDF together with our responses.

Round 2

Reviewer 1 Report

The authors had addressed all the comments and had done the corrections accordingly.

Author Response

The authors would like to thank Reviewer 1 for the careful review and most valuable comments that helped in improving the quality of the manuscript. We have carefully evaluated these comments, now enumerated below together with our responses.

Reviewer 2 Report

To my mind, authors should place the axises XYZ (need to show the coordinate axes) in Figures 7a, 8a, 18a, and 19a. In this case, an explanation of the bending radius of the antenna along the x-axis or y-axis will be more clear.

The text in lines 259-265 is hardly understandable. Authors should rephrase this text. From this part of the text, one can conclude that the bottom patch plays a key role in making the antenna resonant at 2.45 GHz, while the top patches play a major role in making the antenna resonant at 5.8 GHz. Is it true?

Authors should keep the uniformity of writing formulas (5) and (6). 

Author Response

(The authors gave the same response as above.)
